# mPFC spindle cycles organize sparse thalamic activation and recently active CA1 cells during non-REM sleep

Carmen Varela[1,2]*, Matthew A Wilson[1]*

[1]Massachusetts Institute of Technology, Cambridge, United States; [2]Florida Atlantic University, Boca Raton, United States

**Abstract** Sleep oscillations in the neocortex and hippocampus are critical for the integration of new memories into stable generalized representations in neocortex. However, the role of the thalamus in this process is poorly understood. To determine the thalamic contribution to non-REM oscillations (sharp-wave ripples, SWRs; slow/delta; spindles), we recorded units and local field potentials (LFPs) simultaneously in the limbic thalamus, mPFC, and CA1 in rats. We report that the cycles of neocortical spindles provide a key temporal window that coordinates CA1 SWRs with sparse but consistent activation of thalamic units. Thalamic units were phase-locked to delta and spindles in mPFC, and fired at consistent lags with other thalamic units within spindles, while CA1 units that were active during spatial exploration were engaged in SWR-coupled spindles after behavior. The sparse thalamic firing could promote an incremental integration of recently acquired memory traces into neocortical schemas through the interleaved activation of thalamocortical cells.

**\*For correspondence:**
varelac@fau.edu (CV);
mwilson@mit.edu (MAW)

**Competing interests:** The authors declare that no competing interests exist.

## Introduction

Two complementary learning systems acquire and process episodic memories in order to learn adaptive models of the world that generalize beyond specific experiences and continue to incorporate information through the lifetime of an animal (*Marr, 1971*; *McClelland et al., 1995*; *Kumaran et al., 2016*; *Yu et al., 2018*). The hippocampus can form memory traces rapidly; however, over time, interactions between the hippocampus and neocortex are thought to extract statistical regularities from the organism's experiences during wakefulness, and update neocortical models or schemas to increase their adaptive value (*Kumaran et al., 2016*; *Morris, 2006*; *Moscovitch et al., 2016*; *Stickgold and Walker, 2013*). During non-REM sleep, the interactions between the hippocampal and neocortical cell ensembles that underlie these processes are reflected in nested oscillations in the local field potentials (LFPs). The up-states of the slow oscillation (1–4 Hz) provide a processing window in which faster dynamics lead to synaptic plasticity in relevant cell populations. In the neocortex, some cells are preferentially active at the start or at the end of up-states (*Gardner et al., 2013*; *Neske, 2015*), and in the hippocampal CA1 region, sharp-wave ripple oscillations (SWRs, 100–275 Hz) occur primarily during up-states (*Ji and Wilson, 2007*; *Isomura et al., 2006*). Hippocampal SWRs are brief (up to ~100 ms) oscillations that correlate with the reactivation of spatial representations by hippocampal cell ensembles (*Wilson and McNaughton, 1994*; *Davidson et al., 2009*), which are thought to contribute the content in the consolidation of episodic memory (*Buzsáki, 1996*; *Girardeau et al., 2009*; *Ego-Stengel and Wilson, 2010*). Spindle oscillations (6–14 Hz) are coordinated with the slow oscillation and with SWRs (*Siapas and Wilson, 1998*; *Sirota et al., 2003*; *Peyrache et al., 2011*). Spindles are generated in the thalamus when the inhibitory cells of the thalamic reticular nucleus (TRN) evoke inhibitory post-synaptic potentials and rebound bursting in cells of the dorsal thalamus, which then entrain their postsynaptic targets in neocortex (*Steriade et al., 1987*; *Contreras et al., 1997*; *Steriade et al., 1985*). However, the role of

spindles in gating and coordinating communication between hippocampus and neocortex remains to be explored at the cellular and systems level.

The coupling between non-REM oscillations appears to be causally linked to memory consolidation, as suggested by the disruption or enhancement of memory following closed-loop manipulations (*Maingret et al., 2016*; *Latchoumane et al., 2017*; *Ngo et al., 2013*; *Marshall et al., 2006*), and by the association of spindle dysfunction with perturbation of memory function in some neuropsychiatric disorders (*Purcell et al., 2017*; *Manoach and Stickgold, 2019*; *Winer et al., 2019*). However, we are far from understanding the specific mechanisms and single-cell dynamics involved in the hippocampal-neocortical interaction. In particular, information regarding the thalamic involvement is limited. Evidence from anesthetized animals suggests a coordination of thalamic firing with both the neocortical slow oscillation (*Slezia et al., 2011*; *Contreras and Steriade, 1995*) and with hippocampal SWRs (*Lara-Vásquez et al., 2016*), while fMRI results from studies in monkeys indicate an overall decrease in thalamic activity during SWRs (*Logothetis et al., 2012*). A recent study in naturally sleeping rats (*Yang et al., 2019*), found a decrease in activity in cells of the thalamic mediodorsal (MD) nucleus at the time of CA1 SWRs, particularly during SWRs uncoupled from spindles recorded in the skull EEG. This emerging evidence emphasizes the need to study the thalamus, but we have been lacking a multi-region approach (thalamus, neocortex, hippocampus) that would compare in the same preparation the dynamics of cell activity in each region to the sleep oscillations, and that would investigate the dynamics of the thalamic contribution down to the finer timescales relevant for plasticity.

We sought to address this gap by characterizing the fine temporal correlations of thalamic cells with sleep oscillations through extracellular recordings of single units and LFPs in the medial prefrontal cortex (mPFC), hippocampal CA1 region, and nuclei of the midline thalamus in freely behaving rats. We targeted these brain regions because of their anatomical and functional connectivity and their role in memory processes. The limbic thalamus has widespread connections with neocortical regions, and projections from the midline are particularly dense to mPFC (*Vertes et al., 2006*; *Hoover and Vertes, 2007*; *Rubio-Garrido et al., 2009*), a critical region for the recall of consolidated memories (*Maviel et al., 2004*). All the recorded nuclei share circuit, physiological and molecular commonalities (*Guillery and Sherman, 2002*; *Phillips et al., 2019*; *Varela, 2014*). The reuniens nucleus (RE) in the midline sends collaterals to CA1 and mPFC, making it a compelling candidate to coordinate mPFC-CA1 activity (*Hoover and Vertes, 2012*; *Varela et al., 2014*). In addition, although the functional association between RE and hippocampus has received a lot of attention (*Dolleman-van der Weel et al., 2019*), the hippocampus projects to rostral parts of the thalamic reticular nucleus and to prefrontal cortex (*Zikopoulos and Barbas, 2012*; *Cavdar et al., 2008*; *Swanson, 1981*), which could indirectly influence nuclei other than RE. Recording from several midline nuclei, we find evidence for a dual role of the thalamus in hippocampal-neocortical sleep interactions; first, an increase in thalamic firing at the down to up-state transition, which could influence the dynamics of the slow oscillation, and second, the sparse but consistent firing of thalamic cells within individual spindles suggests an interleaved activation of thalamocortical synapses that may promote a flexible integration of recently acquired memory traces into neocortical schemas during systems consolidation.

## Results

We used tetrodes to record extracellularly (single units, LFPs) from the midline thalamus, and from areas involved in recent (CA1) and remote (medial prefrontal cortex, mPFC) memory recall (*Maviel et al., 2004*; *Varela et al., 2016*). During the 1–5 hr recording sessions (in freely behaving rats), the animals remained in a quiet, square-shaped enclosure where they cycled through bouts of sleep and wakefulness; in six sessions from three rats, the animals also explored a radial maze after the recording of sleep, and were brought back to the sleep area after behavior for comparison of sleep activity before and after spatial exploration. LFPs and 333 units were recorded from seven rats; sessions from an additional animal in which no units were recorded were also included for LFP analyses for a total of n = 8 rats (*Figure 1a* displays a subset of tetrode locations in brain sections from one rat; *Figure 1—figure supplements 1–2* document the trajectories in the brain for the electrodes and sessions used for analyses). We selected recording sessions in which there was at least one unit recorded in the thalamus and where reliable detection of sleep events could be performed

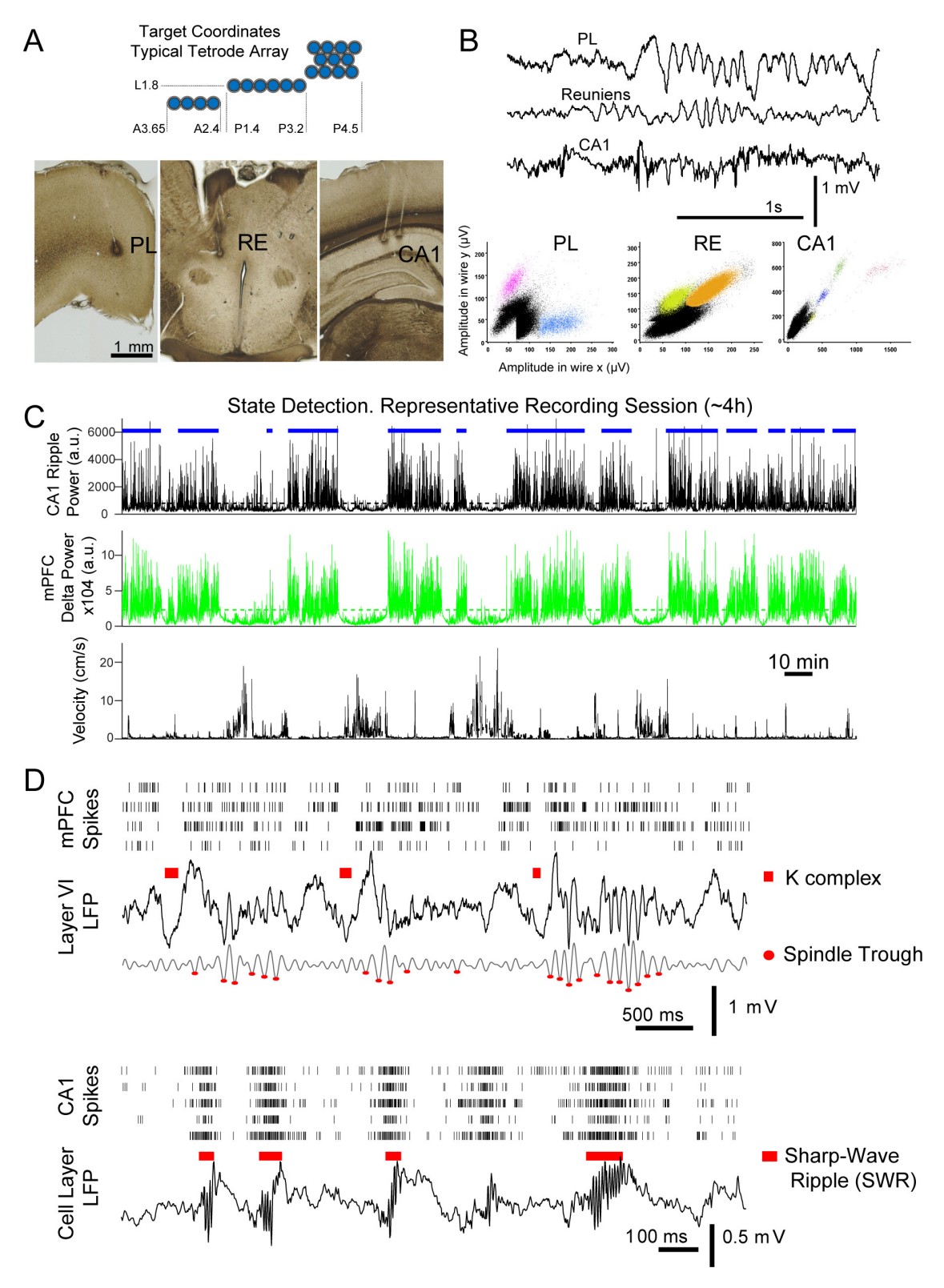

**Figure 1.** Identification of non-REM sleep oscillatory events in mPFC and CA1. (**A**) Diagram of the cross-section of a typical multi-tetrode array and target coordinates. (**B**) Sample LFP traces and units simultaneously recorded in mPFC (prelimbic region, PL), reuniens (RE), and CA1. (**C**) Resting states (blue horizontal bars) were detected from the moving average of the squared filtered LFP, SWR-filtered from CA1 -top panel-, and delta-filtered from mPFC -middle panel-; selected periods used for analyses were confirmed by visual inspection and observation of the corresponding animal velocity -

*Figure 1 continued on next page*

*Figure 1 continued*

bottom panel-. (D) Multi-unit spike rasters (each row shows all spikes from one tetrode) and LFPs, illustrate the detection of sleep events used for analyses; top: detected K-complexes (marked above raw LFP trace) and spindle troughs (marked on the 6–14 Hz filtered LFP) in mPFC; bottom: examples of CA1 sharp-wave ripple detection.

The online version of this article includes the following figure supplement(s) for figure 1:

**Figure supplement 1.** Lesions and electrode trajectory for tetrodes included in analyses.
**Figure supplement 2.** Lesions and electrode trajectory for tetrodes included in analyses (continued).
**Figure supplement 3.** Spindle trough-triggered wavelet scalogram of CA1 and mPFC LFP in sleep compared to quiet and active wakefulness.
**Figure supplement 4.** Event-triggered LFP and spike averages across sessions.

(slow oscillation and spindle activity in mPFC, sharp-wave ripples, SWRs, in CA1; examples of simultaneously recorded LFP traces and units shown in *Figure 1b*). The majority of recordings in the thalamus were from units in the reuniens (RE), and ventromedial (VM) nuclei, and some from the mediodorsal (MD), paratenial (PT), rhomboid (RH) and centromedial (CM; detailed population numbers in *Table 1*). All these nuclei are bi-directionally connected to mPFC, and are functionally related to mPFC and other higher order cognitive regions, including CA1, which also receives direct input from RE (*Dolleman-van der Weel et al., 2019*; *Cavdar et al., 2008*; *Swanson, 1981*). Here, we will use the term limbic thalamus to refer to the thalamic nuclei that we recorded. See the Materials and methods section for target anatomic coordinates, procedures to identify tetrode location, and additional inclusion/exclusion criteria for the analyses reported below.

To study the modulation of unit spikes in the three regions (mPFC, limbic thalamus, CA1) with non-REM sleep oscillations, we first identified times with both high delta (1–4 Hz) power in mPFC and high SWR (100–275 Hz) power in CA1 (as described in Methods); an example of such non-REM sleep state detection for a representative session is shown in *Figure 1c*. Within this state, we detected reference features that reflect non-REM sleep events that are key organizers of memory consolidation: K-complexes (KCs, marked by red bars in *Figure 1d*, mPFC LFP), the troughs of spindle cycles in mPFC (red dots in 1d), and SWRs in CA1 (red bars in *Figure 1d*, CA1 LFP). These LFP events are markers of cell population dynamics, namely, KCs mark the down-states of the neocortical slow oscillation, while SWRs correspond to the high-frequency oscillations that reflect memory-related reactivation of ensembles of CA1 place cells (*Davidson et al., 2009*; *Girardeau et al., 2009*; *Ego-Stengel and Wilson, 2010*; *Wilson and McNaughton, 1994*). Spindles (6–14 Hz) are generated through oscillatory bursting in the thalamic reticular nucleus (TRN), which is thought to then engage cells in the dorsal thalamus and, through them, neocortical networks (*Steriade et al., 1987*; *Steriade et al., 1985*; *Steriade et al., 1993*; *Huguenard and McCormick, 2007*). We considered that the time window of each spindle cycle has a similar duration to that of SWRs (up to ~100 ms) and sought to test if individual spindle cycles represent a window for multi-region coordination; we thus detected individual spindle cycles from the mPFC 6–14 Hz filtered LFP. The detection of individual cycles was also motivated by the occasional observation of few spindle cycles in succession (e.g. only 3–4 cycles following the second detected KC in *Figure 1d*), and by the broader variability in the morphology of the LFP trace in the spindle band compared to the more clearly discrete and consistent signatures of KCs and SWRs (examples in *Figure 1d*). KCs, spindle cycles, and SWRs were detected from the mPFC and CA1 LFPs after applying an amplitude threshold to the squared, filtered trace (1–4 Hz for KCs; 6–14 Hz for spindles; 100–275 Hz for SWRs), at a level of three (spindles) and four (KCs, SWRs) standard deviations above the average LFP voltage across sleep periods in each recording session (see Materials and methods for details). Although the spindle band overlaps with the theta band, we limited the detection of spindle cycles (and KCs and SWRs) to resting states; peri-spindle spectrograms confirmed that detected spindle cycles are functionally different from

**Table 1.** Units recorded in each thalamic nucleus.

| Nucleus | RE | VM | MD | Rh/CM | PT/AM | No histology | Total |
|---|---|---|---|---|---|---|---|
| # of units | 25 | 18 | 7 | 4 | 7 | 2 | 63 |

RE = reuniens; VM = ventromedial; MD = mediodorsal; Rh/CM = rhomboid –centromedian region; PT/AM = paratenial -anterior medial region.

theta (*Figure 1—figure supplement 3*). Peri-event histograms of the LFP and local spiking activity, triggered on the timestamps at the time of largest amplitude of the detected events (trough for KCs and spindles, peak for SWRs), confirmed that the LFP detection identified events that reflect the modulated activity of local cell populations (*Figure 1—figure supplement 4*). Quantifications are reported as mean ± standard deviation (sd), unless otherwise indicated.

*Figure 2a* illustrates the temporal and phase relations between non-REM sleep events, with examples from the raw and filtered mPFC and CA1 LFPs. The overlaid filtered LFP traces (which have been scaled for display purposes), show that increases in SWR amplitude occur preferentially in the positive half-cycle of the delta-filtered LFP, and in the negative halves of the spindle-filtered trace (inset). The average distribution of delta and spindle phases (estimated with the Hilbert transform, see Materials and methods) at the time of SWRs is shown in *Figure 2b* for all sessions. *Figure 2c* shows the peri-event histograms for non-REM detected population events (SWRs, KCs, spindle cycles) across sessions and rats.

Confirming previous results (*Ji and Wilson, 2007*; *Siapas and Wilson, 1998*; *Sirota et al., 2003*; *Peyrache et al., 2011*), we found that KCs, spindles, and SWRs are temporally and phase-coupled. SWRs preferentially occurred following and preceding KCs, phase-locked to the peaks of the delta-

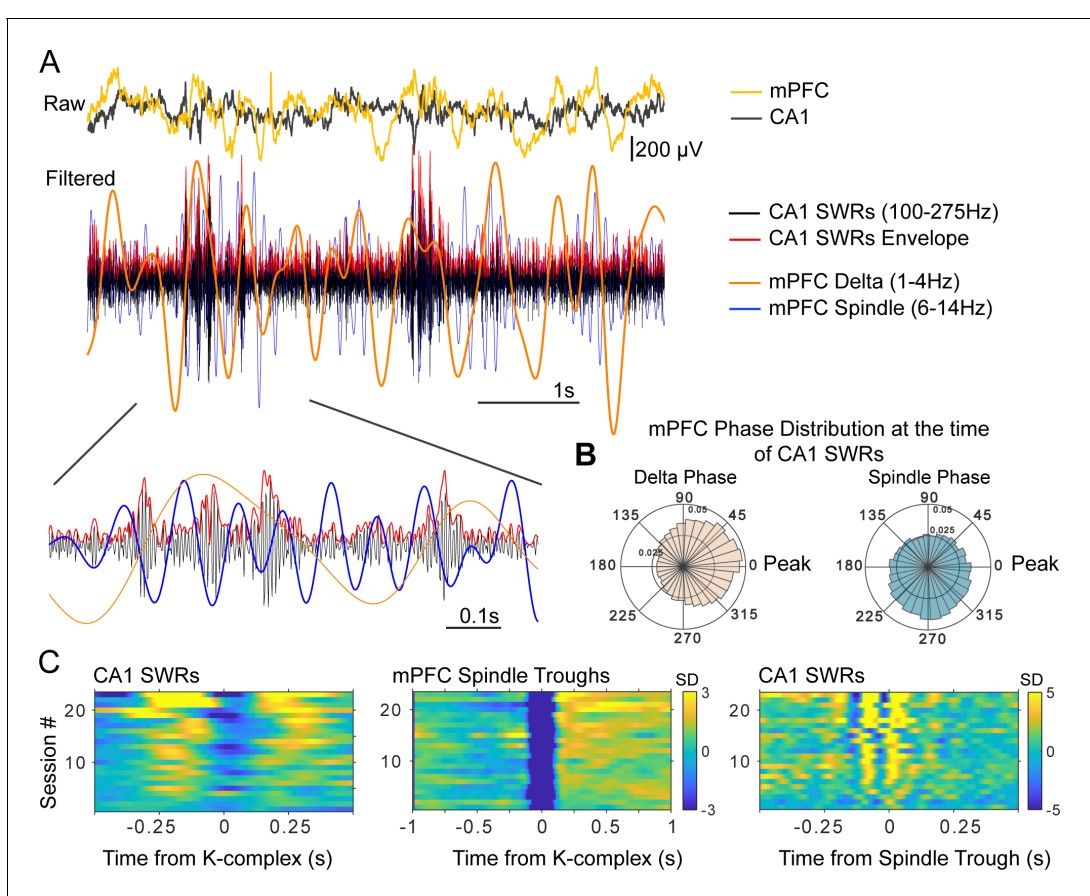

**Figure 2.** Time and phase coordination between non-REM sleep events. (**A**) Overlaid raw and filtered mPFC and CA1 LFPs illustrate the time relations of the relevant oscillatory events (filtered traces have been scaled in amplitude for display purposes). (**B**) Average (across sessions and rats) distribution of delta and spindle phase at the time of SWRs, showing a preferential occurrence of SWRs near the peak of delta and the ascending phase of spindles (inner and outer circle values indicate probabilities). (**C**) Peri-event histograms of detected non-REM sleep events show that SWR probability increases in the ~250 ms windows near KCs (left panel), and are nested in the negative half-cycle of spindles (right), spindles are largest in amplitude following KCs (middle) (color scale bars indicate standard deviations (SD) with respect to a null, shuffled, distribution; scale bar in the middle panel applies also to the left panel).

The online version of this article includes the following figure supplement(s) for figure 2:

**Figure supplement 1.** Phase-Locking of SWRs to mPFC delta and spindle frequency bands.

filtered LFP, and to the troughs and ascending phase of spindle cycles (circular mean and standard deviation of delta phase at the time of SWRs = −0.04 ± 0.43 rad; spindle phase = −1.72 ± 0.93 rad; n = 23 sessions, six rats; *Figure 2—figure supplement 1* shows the delta and spindle phase distributions at the time of SWRs for all sessions included in this analysis).

## Thalamic cells decrease their overall firing and increase bursting with CA1 SWRs

We next investigated the changes in the firing patterns of single units in the limbic thalamus with CA1 SWRs.

Most thalamic units presented a reduced average firing rate during rest compared to wakefulness. In 28 of 32 sessions, the animals (n = 7) alternated between sleep and active wakefulness (in the other four sessions rats were only in wakefulness for less than 5 min and were not included in this comparison), and the firing rate of thalamic units changed from an average of 10.10 ± 4.26 spikes/s during wakefulness to 7.73 ± 3.09 spikes/s during sleep (n = 40 units; p=1.31 × $10^{-6}$, Wilcoxon signed rank; *Figure 3a*).

A decrease in firing rate may be due to the hyperpolarization of thalamic cells. Thalamic cells fire in two modes, burst and tonic, depending on the activation state of a transient conductance that relies on T-type calcium channels (*Perez-Reyes et al., 1998*; *Gutierrez et al., 2001*). Bursts correlate with thalamic hyperpolarization, which is a pre-requisite for T-channel activation, and they can enhance transmission in thalamocortical synapses and induce synaptic plasticity (*Swadlow and Gusev, 2001*; *Leresche and Lambert, 2017*). Therefore, analyzing burst firing can provide clues on the mechanism behind the decrease in firing rate in sleep. Bursting can be detected from extracellular recordings because of its characteristic combination of a relatively long inter-spike interval followed in rapid succession by a few spikes with brief inter-spike intervals (*Lu et al., 1992*), but the relation between thalamic bursts and SWRs has not been explored.

We detected bursts in the thalamus as groups of action potentials with less than 5 ms of inter-spike interval preceded by at least 70 ms with no spikes, a criterion based on the dynamics of the T-channels underlying burst firing in the dorsal thalamus (*Gutierrez et al., 2001*; *Lu et al., 1992*; *Jahnsen and Llinás, 1984a*). With this criterion, we found that 25% of the thalamic cells had at least 5% of their spikes in bursts during wakefulness, while 77.5% did during sleep (average percentage of spikes in bursts during wakefulness 4.60 ± 7.97%, and during sleep 18.65 ± 15.3%; p=1.8 × $10^{-7}$; Wilcoxon signed rank; n = 40 units, seven rats; *Figure 3a*). The increase in bursting was more pronounced around the time of SWRs in CA1 (25.52 ± 17.29% of spikes in bursts in a 2 s window around SWRs; p=1.8 × $10^{-7}$ Kruskall-Wallis; *Figure 3a*). We also estimated thalamic bursting after shifting the timestamps of each SWRs randomly by at least 2 (or 5 s) and calculated the total number of spikes and percent of burst spikes in the same 2 s window around SWRs. In both cases (>2 and>5 s shifts), the average number of thalamic unit spikes was significantly lower in the SWRs windows compared to the shifted timestamps (p<$10^{-3}$), and the percent of spikes in bursts significantly higher (p=0.002 and p<$10^{-3}$ respectively after a 2 s or 5 s shift; Wilcoxon signed rank). Because the T-channels responsible for burst firing need a sustained hyperpolarization for about 50–100 ms to be primed for activation (*Gutierrez et al., 2001*; *Jahnsen and Llinás, 1984b*), the increase in bursting suggests a hyperpolarization of the majority of cells in the limbic thalamus during sleep, and particularly in association with hippocampal SWRs.

Indeed, peri-SWR triggered histograms showed that SWRs occur in a background of decreased thalamic spiking. An example of a peri-SWR histogram for a unit from the RE nucleus is shown in *Figure 3b*, and the peri-SWR for the population of all thalamic units in *Figure 3c*; 55.32% of units in the thalamus showed a reduction in firing of more than two standard deviations (sd) from the null (average −4.23 ± 2.00 sd) in a 200 ms window around the time of SWRs occurrence. Instead, 90.18% of CA1 units increased their firing by 19.85 ± 18.65 sd in a 200 ms window around SWRs (4.91% of CA1 units showed an average decrease of 4.11 ± 1.59 sd), while the mPFC population included 30.95% of units showing an increase (average 6.22 ± 4.32 sd) and 40.48% a reduction in firing (average −3.22 ± 1.40 sd). The peri-SWR histograms for the population of CA1 and mPFC units are shown in *Figure 3—figure supplement 1a*. A few thalamic units (14.89%) showed a pure increase in firing in the 200 ms window around CA1 SWRs; this means an increase in firing larger than two standard deviations from the null distribution and that was not associated with a post-inhibition rebound.

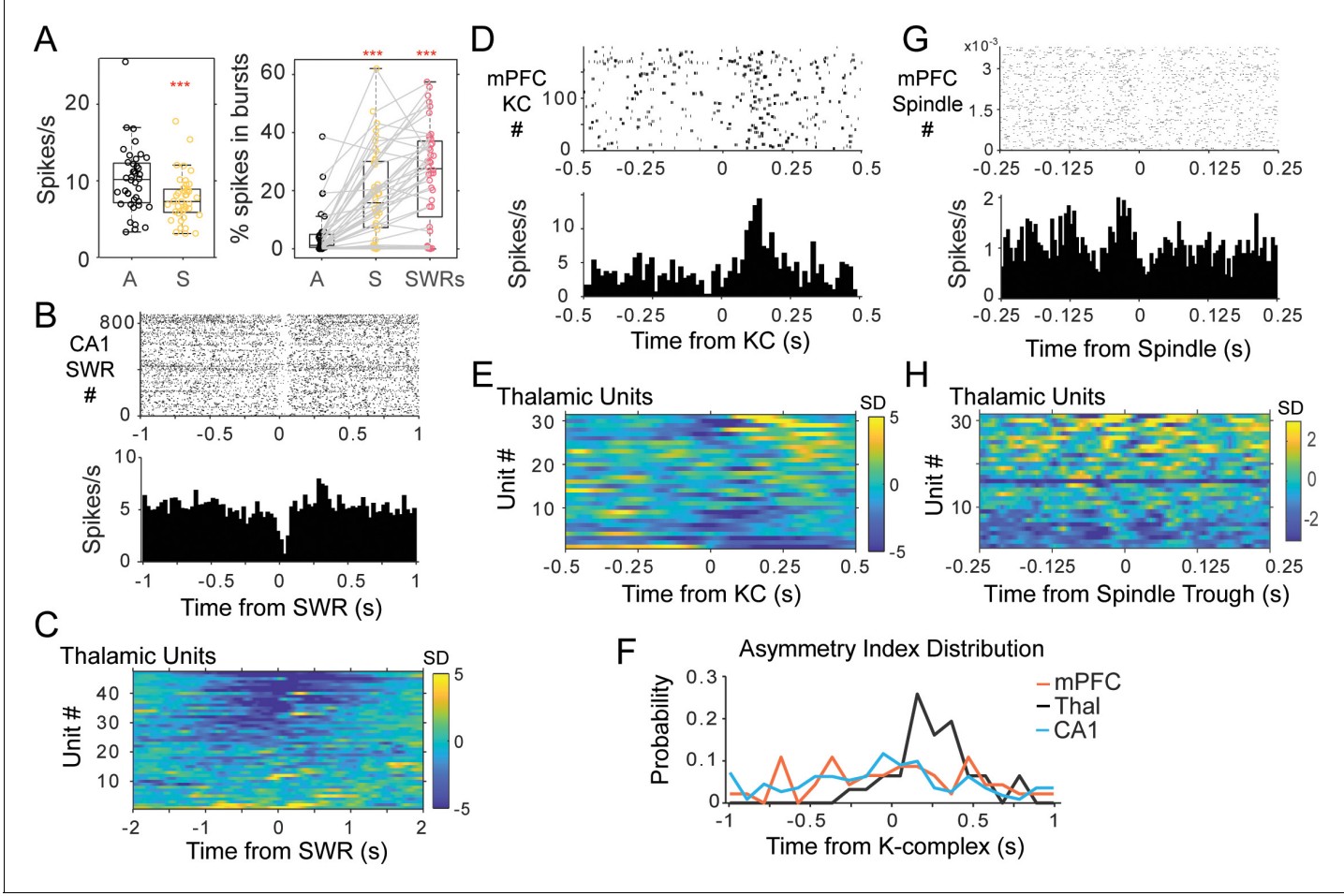

**Figure 3.** Thalamic single unit firing with non-REM sleep oscillations. (**A**) Left, distributions of average firing rate for all units in the awake, A-black, compared to sleep periods, S-yellow. Right, distributions of the percent of spikes in bursts for the same units during wakefulness, sleep, and in a window of 2 s around the occurrence of CA1 SWRs. (**B**) Example of peri-SWR-triggered histogram for a unit recorded in the nucleus reuniens, showing a decrease in firing tightly locked to SWR occurrence (top plot is the raster of unit spikes for all detected SWRs). (**C**) Population results of peri-SWR triggered histogram of spikes for all thalamic units; scale bars units are in standard deviations (SD) from the null, shuffled, distribution. (**D**) peri-KC histogram shows a rebound of firing in a unit recorded in MD thalamus; (**E**) peri-KC histograms for all thalamic units. (**F**) Distribution of the asymmetry index in the three cell populations highlights the asymmetric response in the thalamic population compared to the other two groups. (**G**) Spindle-trough triggered histograms for a unit from MD, (**H**) same for all the thalamic units.

The online version of this article includes the following figure supplement(s) for figure 3:

**Figure supplement 1.** Peri-event histograms of spikes in the three brain regions for all rats and sessions.

Another four units (8.51%) had a rebound in firing (larger than 2sd) that followed soon after a strong decrease in firing (below −2sd) (as the example in *Figure 3b*).

Using a complementary approach, generalized linear models (GLMs) of CA1 activity (SWRs and multi-unit firing rate) with the firing rate in thalamic units as covariate, showed that the firing in CA1 varied inversely with the firing of most thalamic units (61.7%, p<0.001; only one unit in the MD nucleus had a significant positive correlation with SWR rate). For the units that negatively correlated with SWRs, the models predict an average increase of 20.74% in the rate of SWRs for every one spike/s drop in thalamic firing, suggesting anti-correlated activity in the limbic thalamus and CA1 (instead, as expected, the rate of SWRs was positively correlated with the spiking activity recorded in CA1; p value < $5 \times 10^{-189}$ in all electrodes used for SWR detection). The decrease of spiking in thalamic units with SWRs was observed in RE and in other nuclei. Some of the strongest negative correlations with SWRs were observed in RE (unit in *Figure 3b*), and the size of the effect was marginally significant in RE compared to non-RE units (p=0.05). However, there was no difference

between the proportion of SWR-correlated units in RE compared to non-RE units (p=0.159), suggesting that the decrease in thalamic firing with SWRs relates to general excitability changes in the state of thalamocortical networks in the limbic thalamus.

## Temporal correlation of thalamic units with mPFC delta and spindle oscillations

Although we found a decrease in average thalamic firing and an increase in bursting that suggests a hyperpolarized thalamus during SWRs, it is unclear how the remaining thalamic spikes relate to the mPFC delta (1–4 Hz) and spindle (6–14 Hz) oscillation bands, and if there are specific temporal windows with selective increase in thalamic firing. To investigate this, we used two approaches: we calculated firing rates with respect to local events detected in the mPFC LFP (KCs, spindle troughs); and we estimated the phase-preference of unit spikes to the delta and spindle-filtered mPFC LFP.

### Thalamic rebound following the down-state of the mPFC slow oscillation

*Figure 3d* shows the peri-KC histogram for a unit recorded in the mediodorsal (MD) nucleus. Units in the thalamus often fired asymmetrically with respect to KCs, with stronger firing in the few hundreds of milliseconds after the mPFC KC compared to before KCs (average of 2.48 standard deviations at 274.76 ± 135.33 ms after KCs, compared to 1.28 standard deviations at 263.31 ± 145.31 ms before KCs; n = 31 thalamic units; *Figure 3e*; *Figure 3—figure supplement 1b* includes the peri-KC histograms for units in mPFC and CA1). The trough of thalamic firing occurred on average after the mPFC KCs and after the trough of mPFC unit spikes, and coinciding with the trough of SWRs in CA1, suggesting that the neocortical slow oscillation trough leads both the thalamus and CA1 (thalamic unit firing minimum at 10.56 ± 130.75 ms after KCs, n = 31; mPFC units −7.70 ± 72.3 ms, before LFP KCs, n = 46; SWRs 10.33 ± 5.88 ms after KCs).

Previous evidence (*Neske, 2015*) suggests that the thalamus may have a specific role in the initiation of cortical up-states. To assess the specificity of thalamic firing to the down to up-state transition, we compared the firing of thalamic units before and after mPFC KCs. For the population of thalamic cells, 38.71% increased their activity above two sd (from the null distribution) following KCs, that is, in the down to up-state transition, compared to 9.68% of the cells with a significant peak preceding KCs, that is, in the up to down-state transition (19.35 % units had a significant peak both before and after KCs). Instead, CA1 and mPFC activity was more similar preceding and following KCs. SWR probability increased within the 250 ms around mPFC KCs (4.65 sd at 208.15 ± 90 ms before KCs, and 3.02 sd at 266.85 ± 100 ms after KCs; n = 23 sessions). This indicates a delta modulated increase in SWR probability in CA1, with the minimum rate of ripples following briefly after the trough of neocortical activity (10.33 ± 5.88 ms). The modulation of SWRs was paralleled by a modulation of firing in CA1 units (n = 111), which increased their activity both before and after KCs (2.32 ± 1.49 sd at −233.7 ± 146 ms before KCs, and 2.35 ± 1.69 sd at 273 ± 140 ms after KCs); on average, CA1 units presented a trough of activity after KCs (19.53 ± 132 ms). Activity in mPFC (n = 46 units) was modulated in a similar way, with increases in unit firing preceding and following the down-state of the slow oscillation (2.33 standard deviations at −267.30 ± 110.40 ms, before KCs; 2.31 standard deviations at 295.20 ± 120 ms, after KCs).

In other words, while units from the three areas showed no difference in the peak of the histograms in the time window following KCs (p=0.89, Kruskal-Wallis between regions), thalamic units had significantly lower firing before KCs compared to mPFC and CA1 units (p=0.007, Kruskal-Wallis). This suggests that thalamic cells contribute spikes primarily when a new cycle of the slow oscillation starts, while mPFC and CA1 units also fire at up-to-down transitions. We used an additional metric to quantify the differential firing of units in the three brain areas with respect to the time of mPFC KCs. For all the units, we calculated the difference between the maximum firing rate in the 0.5 s windows before and after KCs, normalized to the maximum firing in the two windows. This asymmetry index ranges between −1 and 1, with negative values indicating higher firing before KCs. The distribution of this index (*Figure 3f*) was biased to positive values for thalamic units, but was uniformly distributed for mPFC and CA1 units (average 0.24 ± 0.22 in thalamus; −0.004 ± 0.47 in mPFC; −0.06 ± 0.5 in CA1), consistent with a selective rebound of thalamic firing in the time window following mPFC KCs, relative to thalamic activity preceding the down-state.

## Sparse thalamic firing and consistent activation of cell pairs during spindles

We then analyzed the firing at the timescale of mPFC spindles, and found the strongest modulations in the CA1 multi-unit activity (MUA, all spikes recorded by a given electrode). An example of thalamic unit firing modulation with spindles is shown in *Figure 3g* and the population histograms for the thalamus in *Figure 3h*. Additional peri-spindle-trough histograms for mPFC and CA1 are shown in *Figure 3—figure supplement 1c*. Across the sessions, 84.62% of the CA1 tetrodes showed an increase in spike activity (at least three standard deviations from null distribution in the −100 to 100 ms window of the peri-spindle-trough histogram), with 53.3% and 36% tetrodes showing a similar result in mPFC and thalamus. The increase was less clear and consistent in individual units, even when the local MUA had a clear modulation, which suggests a functional coordination at the cell population level with sparse engagement of individual cells. Spindle oscillations can vary in frequency (*Gardner et al., 2013*) and individual units may not contribute spikes in every spindle cycle, flattening the average correlograms. To investigate the oscillatory activity of thalamic units directly, we looked for peaks in the autocorrelation of single unit spikes.

The autocorrelations of thalamic units did not present strong peaks at the time intervals of spindle frequency, that is, in the time window between 50 and 150 ms, suggesting that the cells did not fire rhythmically in that frequency. *Figure 4a* illustrates the sparsity of thalamic cell firing during mPFC spindles; the panel shows a trace of mPFC LFP with several cycles of spindle oscillation (6–14 Hz filtered trace overlaid in black); the MUA recorded by one tetrode in the thalamus is shown in orange and the spikes of a single unit in magenta. The auto-correlograms next to the raw data were calculated with the spikes fired during sleep, and show the lack of spindle rhythmicity in the single unit (also when calculated with only burst spikes), in spite of the MUA modulation (which can also be seen in the spike raster). Similar results were obtained for all thalamic units, regardless of the nucleus, and including the units that showed the strongest rebound following KCs. The observation of peaks in the autocorrelation of the MUA was not just due to a larger amount of spikes compared to individual units; calculating the MUA autocorrelation in windows with smaller numbers of spikes (equal to the median number of spikes in units) preserved the shape of the MUA autocorrelation (*Figure 4—figure supplement 1a–e*). We calculated the time at the autocorrelation maximum (after the minimum that follows lag 0) and plotted the distribution for unit spikes and for MUA (all the spikes in sleep or a subsample matching the number of spikes of the sparsest unit recorded in the same session; *Figure 4—figure supplement 1f*). We found that the peak of the MUA autocorrelation occurs on average at 119.30 ± 78 ms, whereas the interval is longer for the units, 223.60 ± 113.30 ms (p<0.001, Wilcoxon Mann-Whitney). The peak of the autocorrelation occurred within the spindle band in 63.3% of the curves calculated with MUA (with subsampled spikes matched to the units) and in 17.02% of the units (*Figure 4—figure supplement 1g* shows the autocorrelation of units in which the maximum autocorrelation value falls within the lags that correspond to the spindle band, 6–14 Hz). The autocorrelations with peak values within the spindle band had significantly higher average correlation values for MUA spikes compared to units (MUA, 0.24 ± 0.20; units, 0.05 ± 0.01; p=0.0038, Wilcoxon Mann-Whitney). One thalamic unit that had a strong indication of sustained oscillatory activity showed a peak of the autocorrelation at 140 ms, or ~7 Hz, on the lowest range of the spindle band (*Figure 4—figure supplement 1h*).

Because cells could be firing rhythmically only when they are bursting, we also calculated autocorrelations with only burst spikes, as well as only for the spikes that occurred in the 0.5 s time window immediately following KCs (when spindles are largest). Thus, these analyses were more restricted to conditions in which rhythmicity in the spindle band would be expected to be strongest, and still failed to reveal evidence of sustained rhythmic firing in this sample of thalamic cells. Overall, the results suggest that most thalamic units are not strongly oscillatory and that it is the combined spikes from multiple units that underlie spindle frequency oscillations.

The reduced rhythmicity in the thalamic units compared to the modulation of the MUA, raise the possibility that different groups of thalamic cells contribute to activity in different spindle cycles. A prediction from this hypothesis is that pairs of units should be more likely to be correlated within brief timescales than individual units with themselves. We tested this by calculating cross-correlograms of sleep spikes in sessions in which we recorded more than one unit in the thalamus (n = 39 pairs, four rats); 33% of the pairs had a peak above three standard deviations from the average cross-correlogram (average peak z-score 2.40 ± 1.21 sd). An example is shown in *Figure 4b*, which

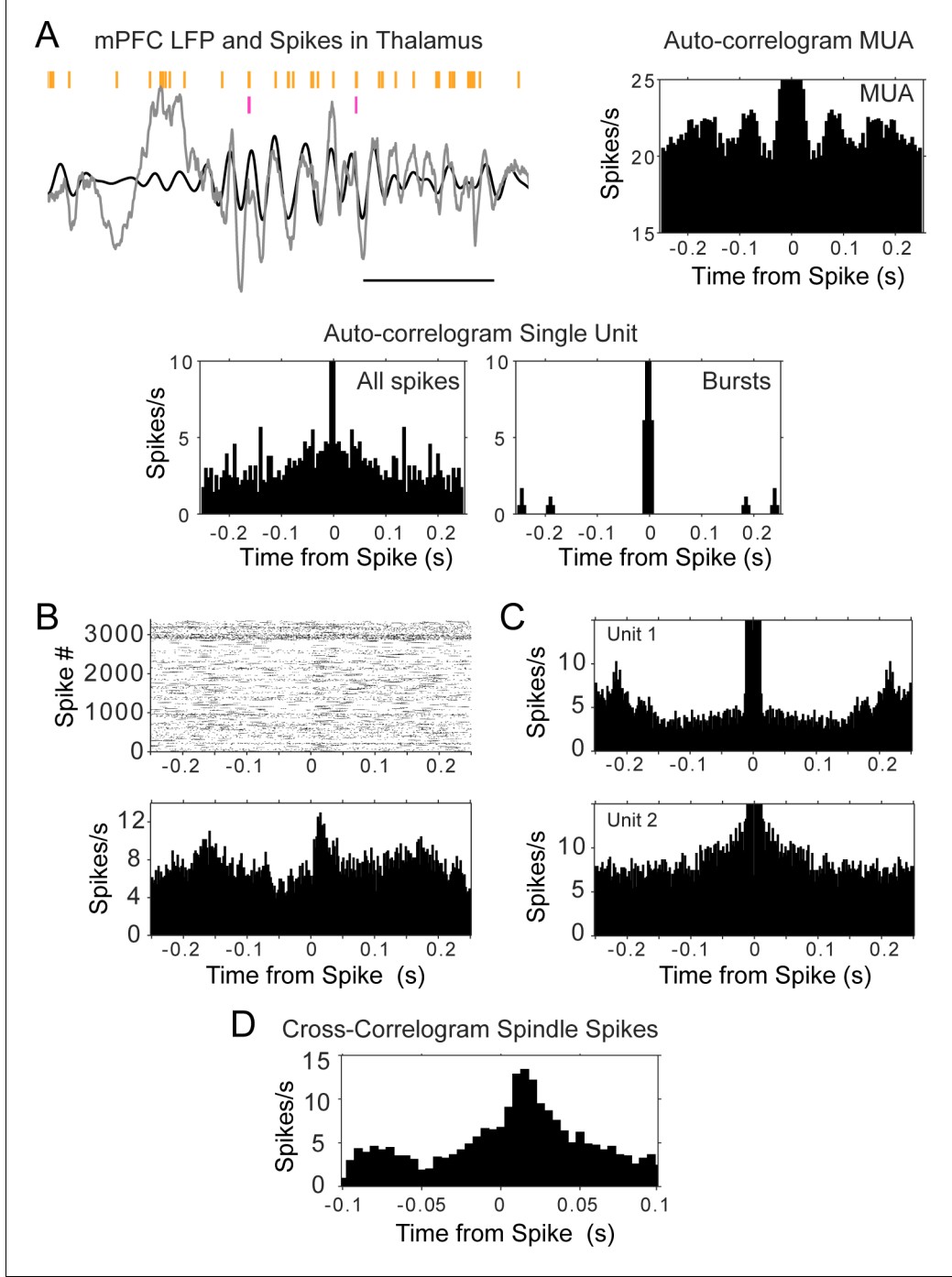

**Figure 4.** Auto- and cross-correlograms of thalamic spikes show lack of single unit rhythmicity but consistent activation of MUA and unit pairs. (**A**) Top left, raw trace of mPFC LFP in gray, with overlaid filtered trace (6–14 Hz, black); spike raster in orange corresponds to all spikes (multi-unit activity, MUA) recorded by one thalamic tetrode, raster in magenta is from a single unit recorded by the same tetrode; scale bar 500 ms. Bottom, auto-correlograms of thalamic spikes during sleep for the single unit (all spikes were used for the plot on the left, burst spikes only for the plot on the right); top right, only the auto-correlogram for the MUA recorded by the same electrode displays rhythmic firing at spindle frequency. (**B**) Cross-correlogram between the sleep spikes of two thalamic units, showing increased modulation within about 50 ms, even when the auto-correlograms of the spikes from each individual unit (**C**) does not show rhythmic firing. (**D**) Cross-correlogram of the same thalamic pair, calculated only with spikes within spindle cycles, suggests spindle-related co-activation.

The online version of this article includes the following figure supplement(s) for figure 4:

*Figure 4 continued on next page*

*Figure 4 continued*

**Figure supplement 1.** Autocorrelation of thalamic MUA and unit spikes during sleep.
**Figure supplement 2.** Cross-correlograms of sleep spikes for simultaneously recorded thalamic units.

displays the cross-correlogram of two thalamic units during sleep; the plots show that the spikes of 'unit 1' follow reliably in a window of ~50 ms after those of 'unit 2' (reference), while neither of the units show a strong auto-correlogram in that period (*Figure 4c*).

All the pairs that showed a peak in the sleep cross-correlograms were correlated in a time scale consistent with the duration of spindle cycles (*Figure 4b*; *Figure 4—figure supplement 2a* for all pair cross-correlograms). We then identified the start and end of spindle cycles by finding the peak of the voltage trace before and after detected spindle troughs. The cross-correlograms between pairs of units showed similar values when calculated only with the spikes that occurred within spindle cycles (*Figure 4d*; *Figure 4—figure supplement 2b* for all pairs); we calculated additional cross-correlograms selecting the unit spikes that occurred during periods of wakefulness as well as during sleep periods between detected spindles (where no spindles were detected for at least 5 s). The strongest cross-correlations were observed during sleep spindles, suggesting sleep specific co-activation of thalamocortical networks (*Figure 4—figure supplement 2b–d*) and providing further support to the hypothesis of a co-activation of sparsely synchronous thalamic cells within spindles (average peak cross-correlograms with spindle spikes $2.74 \pm 0.85$ sd compared to $2.31 \pm 1.18$ sd with sleep spikes that occurred between detected spindles; $p=0.015$, Wilcoxon signed rank test).

The coordinated activity of cell populations drives amplitude and phase dynamics in the local field potentials (*Buzsáki et al., 2012*), and phase coordination has been suggested to underlie routing mechanisms within and across brain regions (*Colgin, 2011*; *Hasselmo and Stern, 2014*; *Wilson et al., 2015*). Therefore, to confirm a functional relation between thalamic spikes and mPFC non-REM oscillations, we assessed the phase locking of thalamic units to the delta and spindle frequency bands of the mPFC LFP.

## Phase-locking of thalamic units to delta and spindle oscillations

We studied the phase-locking of thalamic units to the delta (1–4 Hz) and spindle (6–14 Hz) frequency bands by estimating the LFP phase using two complementary methods, the analytical signal and linear interpolation between consecutive troughs of the spindle and delta filtered LFPs (see Materials and methods for details). *Figure 5a* shows the estimated phase obtained with both methods for the spindle trace in *Figure 4a*. The unit fires sparsely (raster above the trace), but the spikes are likely to occur on the descending phase of the spindle-filtered and on the ascending phase of delta-filtered LFP, shown by the phase histograms calculated for the whole sleep session (*Figure 5b*). We found similar preferential firing for the population of thalamic cells. The phase histograms were fitted for all units to a circular Gaussian (von Mises distribution), with mean and concentration parameters that indicate the direction and depth of the modulation or phase-locking. In order to prevent sampling bias due to non-uniform baseline phase distributions (e.g. due to non-sinusoidal LFP morphology), the unit phase distributions were also normalized to the baseline phase distributions during sleep (*Siapas et al., 2005*).

We found that the majority of thalamic units (67.74% of n = 31 units) showed preferential firing at the descending phases of the spindle-filtered mPFC LFP, as suggested by the positive values of the mean phase distribution (circular mean $0.77 \pm 1.15$ rad; $p<0.05$, Rayleigh test; *Figure 5c*). Thalamic units were phase-locked to the mPFC delta band, closer to the positive peak of the oscillation (64.52%, $p<0.05$, Rayleigh test; circular mean $0.30 \pm 1.04$ rad). The mean preferred phases were robust to the number of spikes; the circular mean was not different when the histograms were calculated with 10% of the spikes ($p=0.68$ for spindle and 0.07 for delta; Wilcoxon signed rank). The amplitude of the thalamic rebound after KCs (reported above) had a tendency to correlate with a preference for earlier phases of delta; although the correlation was not significant ($p=0.08$, r = −0.32), we did not observe such a trend when we regressed mean phase preference to the cell firing rate in a time window of the same length but before KCs ($p=0.65$, r = 0.08).

Interestingly, we found that the phase lags between the spikes of unit pairs that had the strongest cross-correlograms were more consistent than the phase lags between uncorrelated pairs. In the

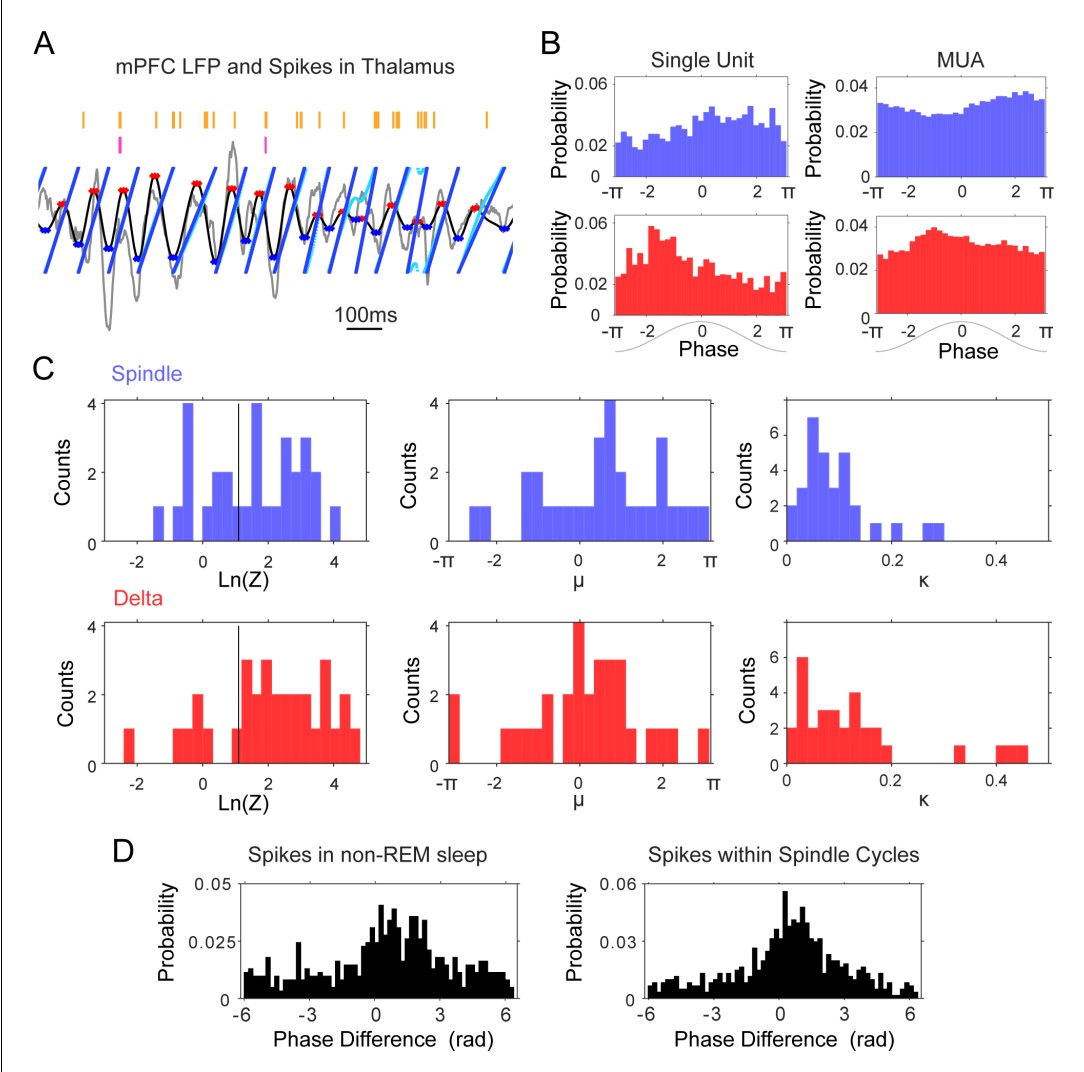

**Figure 5.** Phase-locking of thalamic units to mPFC delta (1–4 Hz) and spindle (6–14 Hz) frequencies. (A) Instantaneous phase estimate for a filtered mPFC LFP trace using the Hilbert transform (cyan) or peak-trough interpolation (blue). (B) Spindle (blue) and delta (red) phase-distribution at the time of single unit and multi-unit spikes recorded during sleep in one tetrode in thalamus (sample rasters in *Figure 7a*); the non-uniform distributions suggest phase-locking of these spikes to the descending phase of the spindle oscillation and the ascending phase of delta. (C) Left, distribution of log-transformed Rayleigh's Z for all thalamic units, used to evaluate the significance of phase-locking to the two frequency bands; values to the right of the black vertical line are significant at the p=0.05 level. Panels on the right side show the distribution of the μ (mean direction, middle panel) and κ (concentration, right panel) of the phase-distribution for all units, which suggest preferential firing at the peak of delta and descending phase of spindle oscillations. (D) Phase lag distributions for the thalamic pair in *Figure 4B–D*; spindle phase-difference between the closest spikes from the two units was calculated for all spikes in non-REM sleep (Left) and for all spikes within spindle cycles (Right), and suggests that the two units are most likely to fire in sequence within about a quarter of a spindle cycle from each other.

The online version of this article includes the following figure supplement(s) for figure 5:

**Figure supplement 1.** Distribution of phase lags between the spikes of simultaneously recorded thalamic unit pairs suggest their consistent activation within spindle cycles.

sessions with thalamic pairs (n = 39 pairs), we calculated the smallest difference between the phase of the spikes during non-REM sleep, and between the phase of the spikes in spindle cycles. The phase difference would be expected to be distributed between – 2π and 2π if units fire within one cycle of each other (positive or negative depending on their activation order), and units that are functionally locked are expected to show a more consistent, less variable phase relation. *Figure 5d* shows the phase lag distribution for the pair described in *Figure 4*, calculated both from all non-REM spikes (left panel) and from the spikes that occurred within spindle cycles (right panel); the

distributions suggest that most of the spikes occur within less than half a cycle of each other; the slight asymmetry with respect to zero suggests the consistent firing of one of the units before the other. We found similar results (*Figure 5—figure supplement 1*) for the thalamic pairs that showed significant peaks in their cross-correlogram. The standard deviation of the distribution of phase lags was significantly lower for units with high spike correlations during sleep (pair cross-correlograms >3 sd) compared to the units that were not temporally correlated. The standard deviation of phase lag distributions within spindles was significantly lower for correlated pairs compared to uncorrelated pairs (2.15 compared to 2.50; p<0.001, Mann-Whitney U-test). These results demonstrate the phase-locked firing of thalamic cells within the period of the spindle oscillation.

Overall, the phase-locking results are consistent with the peri-event histogram analyses in suggesting coordinated thalamic firing with the neocortical delta and spindle oscillations, and demonstrate the phase coordination of pairs of thalamic units within individual spindle cycles.

## Spindle activation of thalamic units modulated with CA1 SWRs

Because SWRs were also locked to preferential phases of the spindle oscillation (*Figure 2b* and *Figure 2—figure supplement 1*), we investigated the association between thalamus and SWRs within the timescale of spindle cycles. We calculated the ripple power within each spindle cycle, and classified cycles in two groups based on the median SWR power (detected spindle cycles and feature distributions for all sessions in *Figure 6—figure supplement 1*). Cycles with ripple power above the median were slightly shorter in duration (102.17 ± 2.01 ms compared to 103.22 ± 2.35 ms when ripple power was low; p<0.05 in 65% of the sessions, although not significant for the mean duration across sessions, p=0.11; Mann-Whitney U-test) and larger in amplitude (229.21 ± 102.14 μV versus 219.47 ± 94.32 μV; p<0.05 in 85% of sessions, p=0.54 across sessions; Mann-Whitney U-test).

*Figure 6* displays the modulation of firing in one of the thalamic units with the strongest spindle modulation in our sample; the spindle-modulated spiking in this unit was enhanced with high SWR power in CA1. The peri-spindle-trough histogram of spikes is shown for all the spindle cycles in the session on the left panel (*Figure 6a*), and for the two functionally defined types of spindle cycles, with low and high SWR power (below or above the median for all spindles), in the middle and right panels. The spindle-trough-triggered histogram of CA1 SWRs for the same session is shown in blue, and confirms that the spindle classification identifies cycles with low or high probability of SWRs in CA1. Because the peri-spindle cycle histograms showed a pattern of both increase and decrease in the spike rate within the timeframe of spindle cycles, we calculated a modulation index as the difference between peak to trough firing relative to the mean in a 200 ms window centered at 0 on the peri-spindle-trough histogram. This index increased in the majority of thalamic units in spindle cycles with ripple power above the median (64.52%, n = 31). In some of the units, like the example in *Figure 6*, we observed a strong phase offset between SWRs and unit spikes (*Figure 6b*); however, this was not the case in all the units, and some had spindle phase preferences that overlapped with those of SWRs recorded in the same session. Likewise, in spite of an increase of modulation index with ripple power, units lacked specific increases of firing at the timescale of SWRs (*Figure 6c* shows the peri-SWR histograms for the same unit as in 6a-b). In addition, a subset of units (22.58%, n = 31) showed stronger modulation with low ripple power spindles, with modulation values that were at least as high as the average for cells that increased their modulation with high ripple power.

The changes in firing modulation with ripple power in spindles could result from associated changes in spindle amplitude, or from a synergistic effect between spindles and ripples. We compared the peri-spindle cycle modulation index for spindle cycles classified based on spindle amplitude or on ripple power (above or below the median). The absolute index values were similar in spindles classified based on spindle amplitude or ripple power (p>0.3, Wilcoxon signed rank), but increased significantly from low to high amplitude spindles (p=0.005) compared to low to high ripple power spindles (p=0.25). The modulation also increased, although less, when comparing spindles with low amplitude and low ripple power to spindles with high amplitude and high ripple power (p=0.014). This suggests that the strongest firing modulation is due to changes in spindle amplitude compared to ripple power or to the combined effect, as expected from the role of thalamic cells in spindle generation (modulation index distributions for the population in *Figure 6d*). Although the modulation by ripple power is not as strong, it could provide an additional boost to the spindle related modulation. To find out if there are synergistic effects between spindle amplitude and ripple power, we detected the spindle cycles with the highest amplitude (upper 25% quartile) and

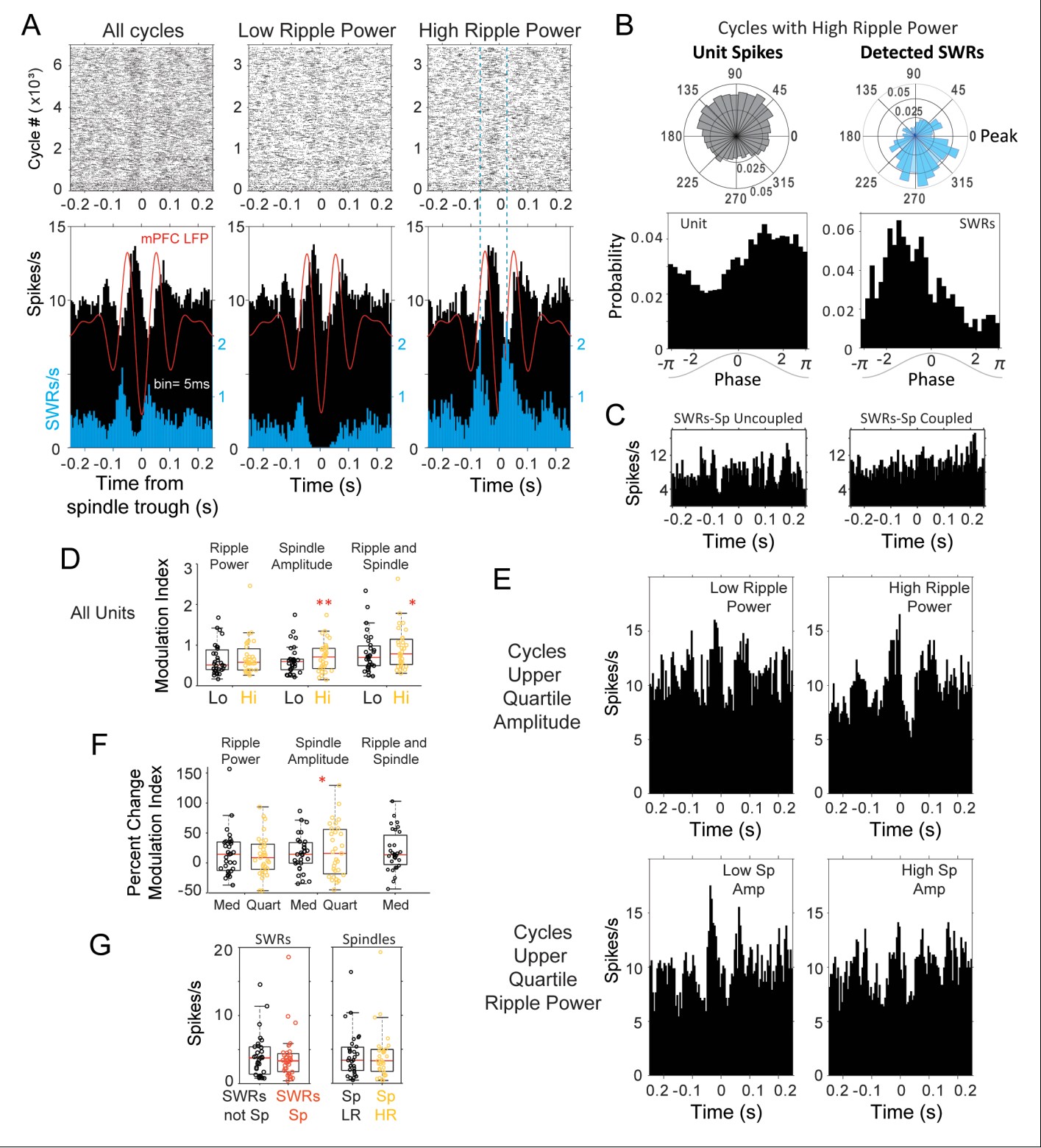

**Figure 6.** Modulation of spindle-related thalamic activity with SWRs. (**A**) Spike rasters and peri-spindle-trough histograms of thalamic spikes from one unit and for SWRs recorded in the same session during sleep. Spindle cycles were classified into low and high ripple power, according to the median power in the CA1 ripple band during the cycle. In this unit there was an increased modulation of firing in spindles with high ripple power (right plot), compared to those with low power (middle); the spindle-triggered histogram of SWRs (blue) confirms the increased probability of SWR events in high-ripple power spindles. The average spindle-triggered mPFC LFP is overlaid in red. (**B**) Example of phase-locking for the unit and SWRs; spikes from the

*Figure 6 continued on next page*

*Figure 6 continued*

thalamic unit in (**A**) occurred preferentially in the descending phase of spindles with high ripple power, in contrast to the SWR preference for ascending phases. (**C**) Peri-SWR histograms of the same unit in A-B for SWRs coupled and uncoupled to spindles showed no clear correlation with SWRs. (**D**) Distributions of spike modulation index and the percent of change in modulation index. (**E**) Increased peri-spindle modulation with high ripple power still observed in spindles that are not significantly different in amplitude (upper amplitude quartile; top row); peri-spindle modulation decreases with high spindle amplitude in this unit (in cycles that are not significantly different in ripple power; bottom row). (**F**) For the thalamic population; changes in spike modulation were significant at the population level with spindle amplitude (each circle one unit; Hi = cycles with high ripple power, Lo = low ripple power; Med = spindles classified based on median ripple power or amplitude, Quart = spindles classified based on upper quartile of ripple power or amplitude). (**G**) Left, average thalamic firing in peri-SWRs histograms with SWRs nested in spindles (SWRs Sp, red) compared to those outside spindles (SWRs not Sp). Right, average firing in spindles with low (Sp LR, yellow) and high ripple power (Sp HR).

The online version of this article includes the following figure supplement(s) for figure 6:

**Figure supplement 1.** Detected spindle cycles for all sessions, classified by mean CA1 ripple power within the duration of the cycle.

**Figure supplement 2.** Additional examples of thalamic units with increased and decreased firing modulation with CA1 ripple power and spindle amplitude.

**Figure supplement 3.** Histograms of mPFC spike activity in 800 ms windows around SWRs coupled and uncoupled to spindles.

compared the modulation of thalamic cells when ripple power was high or low (above or below median). We run a complementary analysis in which we detected spindle cycles with the highest ripple power (upper 25%) and compared the modulation of thalamic cells in cycles with low or high amplitude. *Figure 6e* illustrates an increased modulation with ripple power in the same thalamic unit from panels 6a-c. The peri-spindle histogram for spindle cycles in the upper quartile of the distribution of peak-to-trough amplitudes is split between those that have low or high ripple power; the two groups of spindles are not significantly different in amplitude (p=0.45), yet the unit increases its firing modulation with high-ripple-power spindles (top plots); the same unit shows reduced modulation with large spindle amplitude for spindles with the highest ripple power (bottom). *Figure 6—figure supplement 2* includes additional examples of units that increase and decrease their spindle modulation with ripple power and spindle amplitude. In the population of cells, we did not find evidence of a significant effect of ripple power on the modulation value of thalamic firing in high amplitude spindles (p=0.94; Wilcoxon signed rank), while higher spindle amplitude led to higher modulation values on high ripple power spindles (p=0.04). However, although the modulation values were not significantly different, a majority (61.29%) of units showed a further increase in modulation index with ripple power in high amplitude spindles (average increase 18.15 ± 21.94%); instead, 32.26% of the units increased their modulation with spindle amplitude (in the spindles with highest ripple power; average increase 30.00 ± 20.97%), suggesting that when spindle amplitude is high, ripple power can enhance the modulation of thalamic cells further. The percent change in the modulation index with ripple power and spindle amplitude were positively correlated (Spearman correlation 0.51; p=0.003).

If there is a functional relation with spindle amplitude or ripple power, spindles in the upper quartiles should lead to stronger modulation. We then calculated the fraction of thalamic units in which the increase in modulation in spindles within the upper quartile was higher than the increase observed with the median classification. The increase in modulation index was higher in the upper quartile of spindle amplitude in 29% of thalamic units and with ripple power in 22.6% units. The absolute modulation values were higher for spindle amplitude compared to ripple power and positively correlated (average of 1.16 ± 0.57 in the upper quartile of spindle amplitude compared to 0.60 ± 0.26 in the upper quartile of ripple power; p=0.04, Mann-Whitney); units with stronger modulation with spindle power had higher values of ripple modulation (Spearman coefficient >0.7; p<0.04). For spindle amplitude, the lower modulation values also became lower with the quartile classification, leading to a wider spread of the distribution (when spindles were classified based on quartiles compared to median; p=0.04; F-test) which was not the case with ripple power (p=0.36; F-test; *Figure 6f*). These results also suggest that the strongest firing modulations in thalamic cells are related to the spindle oscillation, and that ripples in CA1 are associated with smaller changes in firing modulation in about 20% of thalamic cells.

Although we observed clear modulations of firing with spindles, the absolute number of spikes in single units did not change significantly with spindles low or high in ripple power, nor with SWRs coupled or uncoupled to spindles (*Figure 6g*).

We also observed a stronger modulation with ripple power in the cross-correlograms between unit pairs. We calculated the modulation index for the cross-correlograms of thalamic pairs and found that it increased by an average of 26.64% in 48.72% of the pairs in spindle cycles with high ripple power and decreased by 20.90% in the rest of the pairs (n = 39). Altogether, the data is consistent with the hypothesis that the up and down modulation of firing of thalamic cells during spindles is dynamically regulated in conjunction with changes in ripple activity in the hippocampus. This could influence spindle-specific thalamocortical synaptic plasticity and facilitate the integration of hippocampal memory traces into neocortical networks.

At broader time scales (seconds), we had found a decrease in thalamic firing and increase in bursting with SWRs (*Figure 3*). However, it is possible that only spindle-nested SWRs are correlated with thalamic units. To test this possibility, we calculated the cross-correlograms of thalamic units with SWRs that occurred within the time window of detected spindle cycles, and compared them to a sample of equal number of SWRs that did not occur within spindle cycles. We compared the mean spikes/s in the 100 ms central window of these cross-correlograms between thalamic single units and SWRs within or outside of spindles. We found low firing rates but no significant difference in thalamic firing near SWRs that occurred within detected spindle cycles (mean firing 3.83 ± 3.49 spikes/s with spindle-nested SWRs, compared to 3.99 ± 3.39; p=0.18, Wilcoxon signed rank; n = 31); only when broadening the window of the cross-correlograms to at least 1 s around SWRs, did we find a slight decrease in thalamic firing with SWRs nested in spindles (mean firing 3.73 ± 3.43 spikes/s with spindle-nested SWRs, compared to 3.94 ± 3.35; p<$10^{-2}$, Wilcoxon signed rank; n = 31).

In mPFC, previous work had shown increased correlation between SWRs and mPFC units outside of spindles. We calculated a peri-SWR histogram of mPFC unit activity with the same parameters used in *Peyrache et al., 2011* (2 ms bins in a window of 400 ms before and after SWRs), and separated the peri-SWR histogram based on ripples that occurred within or outside spindle cycles (using a sample with the same number of coupled and uncoupled SWRs). We z-scored the resulting histograms to the same shuffled distribution (after randomizing all SWR timestamps). The resulting peri-SWR histograms are shown in *Figure 6—figure supplement 3a*; we found that there was no significant difference for the population of mPFC units between SWRs coupled or uncoupled to spindles (z-score 0.17 ± 1.26 for coupled-SWRs and 0.24 ± 0.86; p=0.46, Wilcoxon signed rank). However, the units with the strongest peri-SWRs correlograms (calculated with all SWRs), showed differential firing for SWRs coupled or uncoupled to spindles, such that their firing decreased with SWRs coupled to spindles (units with the strongest negative correlation had more negative values with SWRs uncoupled to spindles). We repeated the peri-SWR histograms for multi-unit spikes (all spikes recorded by one tetrode). With this analysis we found stronger correlation of mPFC spikes with SWRs coupled to spindles (z-score 0.83 ± 1.34 compared to uncoupled, 0.09 ± 1.53; p=0.002). We also found stronger correlations with CA1 firing (55.81 ± 28.40, 50.40 ± 30.51; p<$10$–3; Wilcoxon signed rank), and no significant difference in the thalamic tetrodes (−64 ± 1.42, 0.34 ± 0.95; p=0.17). Overall we find that, at the cell population level, both CA1 and mPFC increase their firing significantly during SWRs coupled to spindles, and that some mPFC cells may be more strongly modulated (up or down) with SWRs that are not coupled to spindles. This may reflect two different dynamics in mPFC, one driven by the spindle oscillation, which may engage cell population activity in coordination with CA1, and one, outside spindles, in which certain mPFC cells present larger changes in firing rate with SWRs.

## Multi-region correlations after spatial exploration

To explore the possibility that awake behavior influences the multi-region coordination and thalamic contribution, we analyzed six sessions in which the rats (n = 3; two sessions from each rat) explored a radial maze (a 4-arm maze in one rat, 8-arm maze in the other two). The animals explored the maze freely for 30 mins, collecting pellet rewards at the end of each arm. In these sessions, we recorded an average of 52 min of sleep before and 78 min of sleep after spatial exploration to compare the non-REM sleep events and their correlation with single units before and after spatial experience. All the sessions showed an increase in the amplitude (peak-to-trough) and rate (per second) of SWRs during sleep after the maze exploration, as well as an increase in the amplitude of KCs (p=0.03, Wilcoxon signed rank), while spindles did not change significantly in number or amplitude (p>0.5). As a control for the possibility that these features change over time, we used another set of four longer recording sessions from two of the rats, in which we had recorded continuously (no maze

exploration) for a similar amount of time. We compared the SWRs amplitude and rate in the first half to the second half of these sessions, finding no difference in amplitude nor rate between the first and second half of the recording (p>0.2); the total amount of sleep time was also not different in the first and second half of the recording, and not different from the pre-behavior or post-behavior sleep epochs in the other sessions with 'run' epochs (p>0.1 in all cases).

Although the number and amplitude of detected spindle cycles did not change, there was a significant increase in CA1 SWR power within spindle cycles (p=0.03, Wilcoxon signed rank), consistent with the overall enhancement of SWR activity in CA1. To find out if there is selective activation of behavior-relevant CA1 cells during spindles, we classified CA1 units recorded in these sessions (n = 42) into those that were active (n = 27) or not (n = 15) when the animal was on the maze (based on visual inspection during manual spike sorting, i.e., unit clusters present both on the maze and during sleep, or only in subsequent sleep; examples from one tetrode in *Figure 7—figure supplement 1a*). These two groups of CA1 units did not differ in firing rate in the sleep post-behavior (calculated as the mean number of spikes over the total sleep time, or as the reciprocal of the mean inter-spike interval; p>0.1, Mann-Whitney U-test). However, we found evidence that units that were active during maze exploration are more involved in SWR-spindle oscillations, as suggested by their higher firing rate and spiking at preferential spindle phases when they participate in SWRs nested in spindles. *Figure 7a–b* shows examples of the spindle temporal and phase organization of SWR spikes for two units active and two not active in run. Units that were active during maze exploration were more likely to be active within spindle cycles with high ripple power than units not active on the maze (p=0.009, Mann-Whitney U-test for the mean number of spikes in spindle cycles and p=0.039 for the percent of spindle cycles with unit spikes; examples in *Figure 7a* and population distributions in 7 c). A two-way ANOVA to compare the effect of the 'RUN or not in RUN' factor and ripple power being 'low or high' on the % of cycles with spikes and the mean number of spikes/spindle cycle found both factors to have a significant effect on the spike metrics (p<0.007 for 'RUN compared to not in RUN'; p<0.01 for 'low versus high ripple power'), with no interaction between the two factors (p>0.15). Pairwise comparisons found that the % of spindle cycles with CA1 unit spikes and the mean number of spikes were significantly higher for the combination of units 'in RUN' and cycles with high ripple power, compared to all other combinations (for the % of spindle cycles with CA1 spikes, p values were = 0.0002 when compared to units not in RUN and cycles with low ripple power; p=0.001 compared to RUN with low ripple power, and p=0.028 compared to non in RUN with high ripple power; p values for mean number of spikes were p=0.0002, 0.0007, 0.017 for the respective comparisons; multiple comparison Tukey-Kramer test).

Likewise, the peri-SWR histograms of CA1 cells that had been active during run showed significantly higher peaks (mean 5.09 ± 4.23 spikes/s), compared to those units that were only active in the sleep post-behavior (mean 2.34 ± 2.09 spikes/s; $p<10^{-2}$, Mann-Whitney U-test; population distributions in *Figure 7d*, left). Although units active in run had higher firing rates within SWRs, we did not find differences between the maximum nor mean firing in SWRs that were nested within spindle cycles compared to a sample of an equal number of SWRs not coupled to spindles (p>0.3 for both units in run and units not active in run; Wilcoxon signed rank). However, the spikes fired by units active in run during SWRs nested in spindles occurred at preferential phases of the spindle oscillation; phase distribution examples are shown in *Figure 7b* for the units in 7a (circular mean = −1.48 ± 1.33; concentration = 0.63 ± 0.32; mean p value = 0.027 ± 0.05, Rayleigh test; n = 27 units). The phase modulation in the same spindle-nested SWRs, was not significant for units not active in run (circular mean = −1.20 ± 1.55; concentration = 0.51 ± 0.29; mean p value = 0.218 ± 0.28; n = 14 units, one unit was removed from this analysis due to low firing in SWRs). We used a two-way ANOVA to compare the effect of factors 'RUN-not in RUN' and 'SWRs coupled-not coupled to spindles' on phase locking, and found that Rayleigh's Z was significantly higher for RUN units and when SWRs were coupled to spindles (p values 0.007 for 'RUN-not in RUN' and 0.015 for 'SWRs coupled-not coupled to spindles') with no interaction between the two factors (p=0.08). A pairwise comparison showed that the combination of units in RUN with spikes occurring in SWRs coupled to spindles had the highest Z values compared to all other cases (p=0.01 when compared to units not in RUN in SWRs coupled to spindles; p=0.0026 compared to units in RUN when not coupled to spindles; p=0.002 for units not in RUN and SWRs not coupled to spindles; multiple comparison Tukey-Kramer test). The distribution of Rayleigh's Z for all conditions is included in *Figure 7d*. These results suggest an enhanced contribution of recently active CA1 units to SWRs,

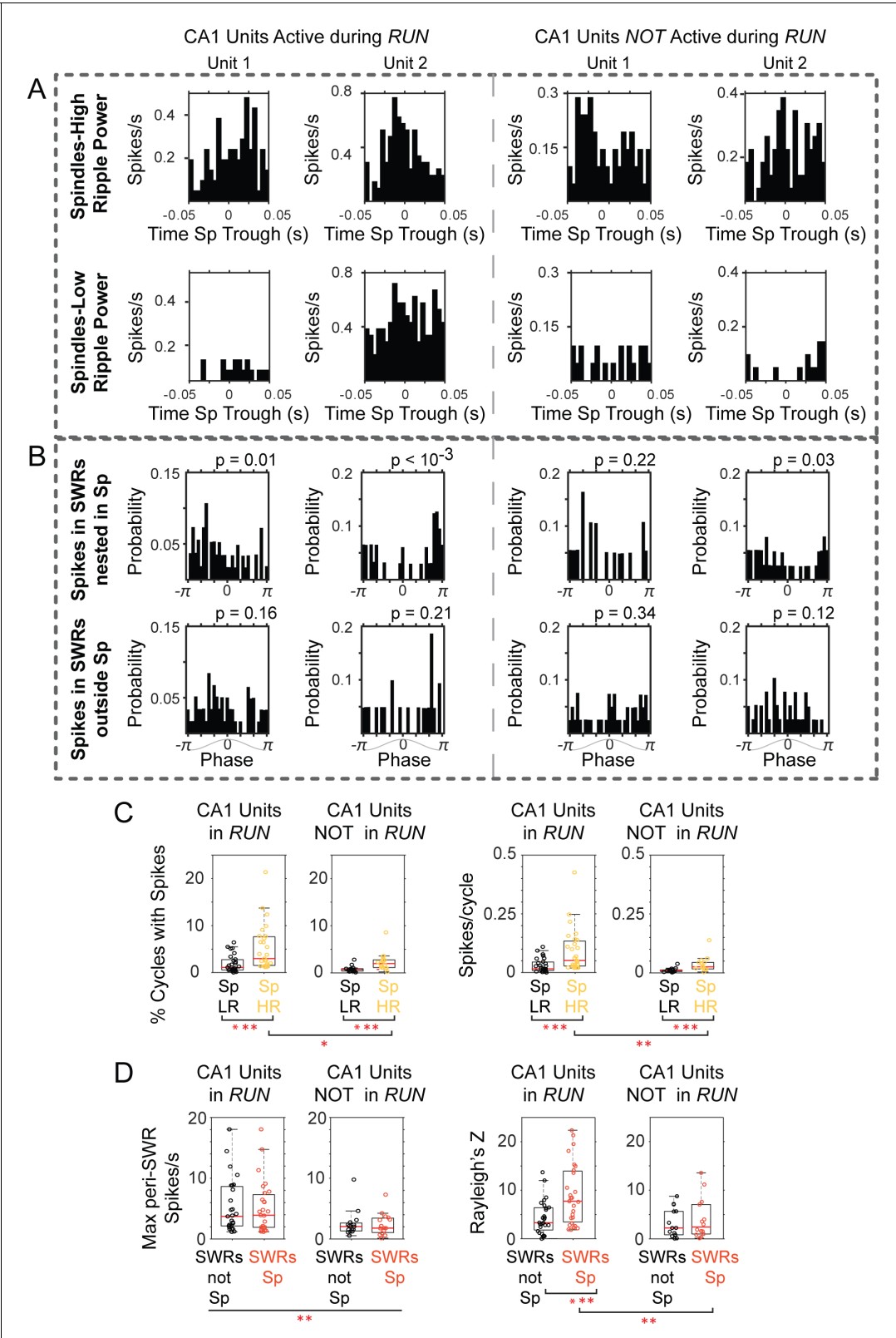

**Figure 7.** CA1 units active during 'run' show stronger activation in spindles with high ripple power, and firing at preferred spindle phases. (**A**) Peri-spindle trough histograms of the spikes of 2 CA1 units active during exploratory behavior (left) and 2 CA1 units recorded by the same tetrode that were only active during the sleep after maze exploration (right); top plots, spindles with high ripple power; bottom plots, spindles with low ripple power (sorted clusters for these units in *Figure 7—figure supplement 1*). (**B**) Distribution of mPFC spindle phases at the time of unit spikes in SWRs nested in

*Figure 7 continued on next page*

*Figure 7 continued*

spindles (top) or outside (bottom); p values on top of the plots from Rayleigh test. (**C**) Population distributions of the percent of spindle cycles in which the CA1 units fired spikes (left panels; Sp LR = spindles with low ripple power in CA1, Sp HR = spindles with high ripple power in CA1), and the mean firing for CA1 units active or not in run (right). (**D**) Left, distribution of peak firing rate in the peri-SWR histograms for CA1 units active or not during run, separated by SWRs nested or not within spindle cycles (SWRs Sp vs. SWRs not Sp respectively); right, distribution of Rayleigh's Z value suggests preferential spindle phase of CA1 units active in run when firing in SWRs nested in spindles. One asterisk indicates p<0.05, two p<0.01, three p<0.001 (non-parametric tests).

The online version of this article includes the following figure supplement(s) for figure 7:

**Figure supplement 1.** Clusters of sorted CA1 units from one tetrode and correlations during maze exploration.

including those SWRs associated with thalamocortical spindles, when the spikes of units active in run are more strongly locked to the spindle oscillation.

We then characterized post-exploration changes in thalamic activity, and did not find significant changes induced by maze exploration in this sample of thalamic units (n = 9). *Figure 8a* shows an example of a thalamic unit with a post-KC rebound, which had a similar profile before and after behavior. Thalamic units also had similar peri-event histograms with mPFC spindle cycles before and after behavior (*Figure 8b*), and no change in spindle phase distribution mean or concentration values (p>0.3, Mann-Whitney U-test); the spindle modulation index increased on average after behavior although not significantly (0.52 ± 0.30 before maze exploration and 0.84 ± 1.03 after; p=0.43 Wilcoxon signed rank test). This is in contrast to an increase in the spindle correlations with SWRs and with CA1 units that were active on the maze. *Figure 8c–d*, shows examples of peri-spindle histograms of CA1 SWRs from the same session in which the thalamic unit in *Figure 8b* was recorded. Although the peri-spindle-trough histograms for the thalamic unit are similar before and after exploration (or even show a slight decrease in modulation), the CA1 unit (which was active during run) and the SWRs show enhanced modulation with spindles post-exploration. On average, we observed a trend for an increase in CA1 unit involvement in spindles after maze exploration. 60% of the CA1 units fired more spikes in high-ripple power spindle cycles after exploration compared to before (p=0.12), and fired in a larger proportion of high-ripple power spindle cycles (p=0.08; Wilcoxon signed rank; distributions for all the CA1 units in *Figure 8e*). A two-way ANOVA (to compare 'Before-After' experience and 'Low-High ripple power' spindles) yielded consistent results, finding a significant difference in engagement in spindles with 'Low-High ripple' power (p=0) and no significant effect for the 'Before-After' factor (p>0.06). It is possible that the lack of significant differences in cell contribution to high-ripple power before and after exploration is due to the low memory demand on this task.

We tested the possibility of selective modulation of thalamic units with spindle cycles that engage CA1 units that were active during maze exploration (compared to spindle cycles with spikes from CA1 units that were not active during behavior). We calculated the peri-spindle histogram of thalamic spikes with spindle cycles containing spikes from the CA1 units that were active on the maze, and compared it to that of spindle cycles with spikes from CA1 units not active during exploration. We found no significant differences between the peak or the average firing values of the cross-correlograms between the two types of spindles and thalamic units (p>0.3, Wilcoxon signed rank), nor any differences between the mean number of thalamic spikes per spindle cycle or the percent of spindle cycles with spikes from thalamic units (p>0.1).

Another possibility is that thalamic cells specifically change their modulation with spindles when CA1 units are reactivating representations of spatial trajectories that were experienced while the animal was on the maze. During exploratory behavior, CA1 cells with neighboring place fields become active as part of theta sequences, showing strong phase-locked activation in the theta band when the animal is running (*Figure 7—figure supplement 1b*); these units then tend to be co-active in resting periods that follow behavior, often during SWRs (*Davidson et al., 2009*; *Wilson and McNaughton, 1994*). We selected pairs of CA1 units that had a clear co-modulation during run (n = 19 CA1 pairs; examples in *Figure 7—figure supplement 1b*), and calculated the peri-spindle histogram of thalamic spikes with spindles in which these pairs were active, compared to cycles in which they were not, and found no difference in thalamic activity (p>0.9). Because the sample of thalamic units in these sessions was small (n = 9), we repeated these analyses with all the background spikes (MUA) recorded by thalamic tetrodes, finding no significant changes (p>0.1).

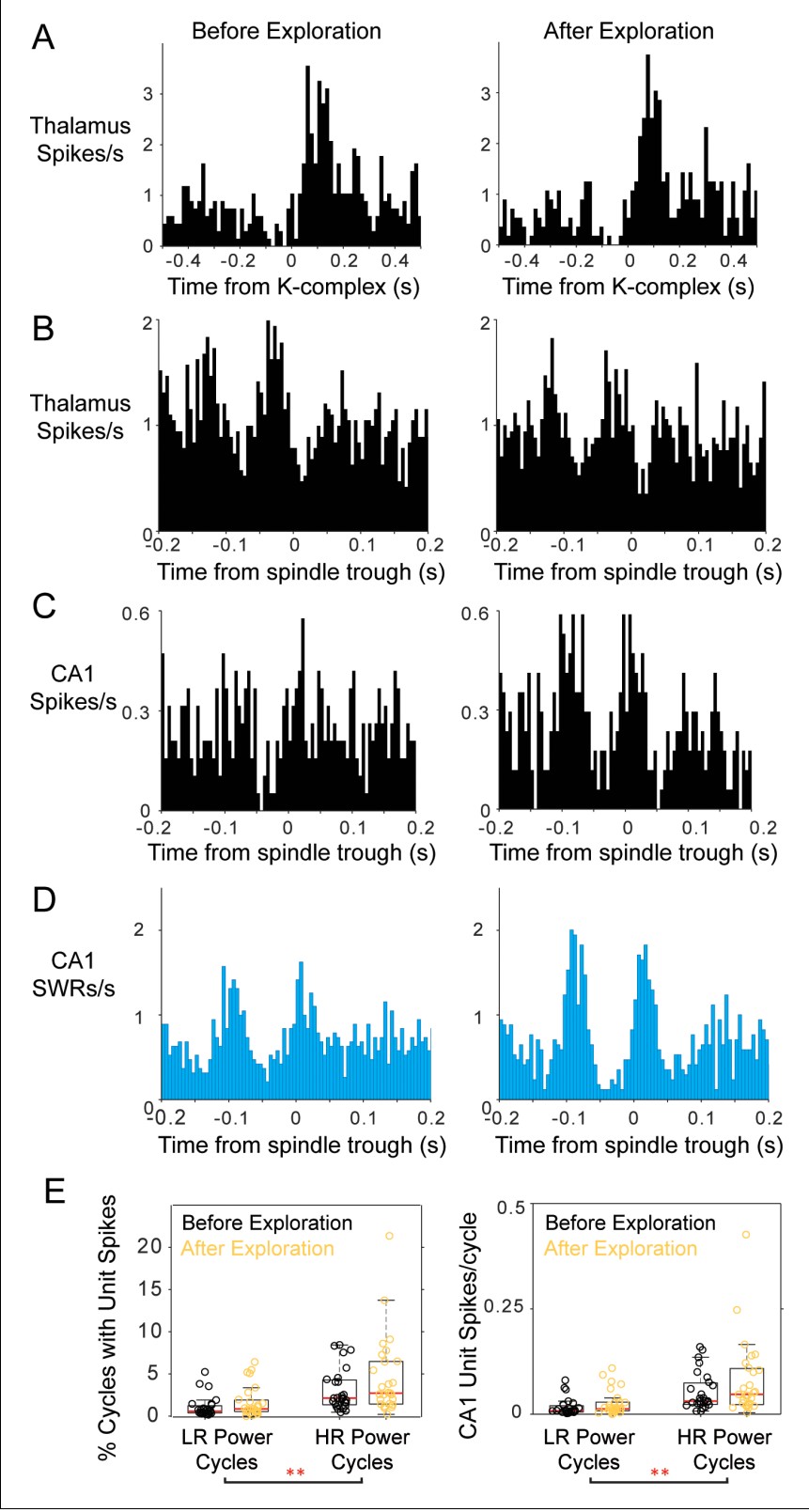

**Figure 8.** Modulation of thalamic and CA1 activity before and after free exploration on an 8-arm radial maze. Peri-event histograms display fairly stable modulation of spiking activity in the thalamus with K-complexes before and after spatial exploration; KC-triggered histograms of thalamic spikes are shown in (**A**) and peri-spindle histograms in (**B**); instead, a CA1 unit recorded in the same session increases its modulation with spindles (**C**) in the sleep

*Figure 8 continued on next page*

*Figure 8 continued*

following behavior. (D) Spindle-triggered SWR histograms show an increase in SWR modulation following behavior. (E) Distributions of the percent of spindle cycles with CA1 unit spikes and the average number of CA1 spikes/spindle cycle before (black) and after exploration (yellow); each circle is one CA1 unit; most CA1 units increased their contribution to spindles post-behavior.

In summary, these analyses suggest that spindles help organize the activation of recently active CA1 units during SWRs but failed to demonstrate a selective modulation of thalamic units with behaviorally relevant CA1 unit activation. It is possible that studying a larger sample of thalamic units, or after behavioral tasks with stronger memory demands, will reveal task-specific modulations between these three brain regions. Nonetheless, these results indicate that there is a spindle related organization of the spiking of task-relevant CA1 cells in spindle-nested SWRs. This could promote CA1-driven synaptic plasticity in mPFC networks during SWR reactivation during spindles.

## Discussion

Here, we report on the temporal correlation between cells in the limbic thalamus and the hallmark oscillations of non-REM sleep. We recorded single units and LFPs from functionally connected regions in the mPFC, CA1, and midline nuclei of the thalamus, including RE, MD and VM nuclei. We found that thalamic cells in these nuclei often reduce their firing and are more bursty with CA1 SWRs during non-REM sleep. Although firing in the thalamus becomes sparse, spikes are correlated with those of other thalamic cells and with LFP population events in mPFC and CA1. Specifically, thalamic units were phase-locked to the delta and spindle frequency bands recorded in deep layers of mPFC, and some thalamic cells showed a strong rebound of activity following the down-state of the mPFC slow oscillation (1–4 Hz). In addition, individual thalamic cells rarely contributed spikes in consecutive cycles of the spindle oscillation, but showed consistent co-activation with other thalamic cells within the period of spindle cycles. We also found that CA1 SWRs were strongly correlated with individual cycles of mPFC spindles, a correlation that was enhanced following spatial exploration. Furthermore, CA1 units that were active during behavior were more strongly activated during SWRs compared to units not active during behavior, and including during SWRs nested in spindle cycles, when they phase-locked to the underlying spindle oscillation. The evidence raises the hypothesis that sparse synchronicity in thalamic cell ensembles facilitates CA1-mPFC consolidation processes during non-REM sleep. The enhancement of ripple-spindle correlations after exploratory behavior and preferential engagement of recently active CA1 units in SWRs and their phase-locking to spindles, further suggest that individual spindle cycles provide time windows in which the interleaved activation of thalamic cells could promote incremental synaptic plasticity and facilitate the integration of recently acquired memory traces into neocortical networks.

### Thalamic inhibition during hippocampal SWRs

The hypothesis of a multi-region interaction that stabilizes and integrates recently acquired episodic memory representations into neocortical schemas is supported by a stream of reports that point to the involvement of the thalamus (*Ji and Wilson, 2007*; *Siapas and Wilson, 1998*; *Sirota et al., 2003*; *Peyrache et al., 2011*; *Latchoumane et al., 2017*; *Logothetis et al., 2012*; *Mölle et al., 2006*; *Staresina et al., 2015*; *Rothschild et al., 2017*). However, the investigation of the role of thalamocortical cells in these processes has lagged behind; evidence from unit recordings in naturally sleeping animals is still limited, and we have been lacking simultaneous recordings of LFPs and single units from these brain regions to study their dynamic modulation within the same animal. Current results point to a negative correlation between thalamic activity and CA1 SWRs, which is suggested by the decrease in fMRI bold signal in the thalamus of monkeys (*Logothetis et al., 2012*) and the reduction of firing in MD and RE thalamic cells with SWRs (*Lara-Vásquez et al., 2016*; *Yang et al., 2019*). Our results extend these reports with the observation of an increase in thalamic bursting, which suggests that inhibition of the thalamus (as opposed to a decrease in excitatory input) explains the negative correlations between hippocampal and thalamic activity. One possibility is that the GABAergic TRN inhibits the thalamus near the time of SWRs. It is also possible that changes in neuromodulatory tone during non-REM sleep could lead to the hyperpolarization of cells in the dorsal

thalamus in coordination with a change in CA1 state that facilitates SWR occurrence. The finding that cells in all the thalamic nuclei we recorded from were coordinated with CA1 and mPFC suggests a non-specific process that may be under neuromodulatory state control, without ruling out the involvement of the TRN. Because RE projects directly to CA1 and to the entorhinal cortex (*Vertes, 2015*), a change in firing mode in RE cells that project to hippocampus could further influence CA1 state transitions and SWRs occurrence.

Thalamic cells with an increase in firing with SWRs were rare in our sample (~15%). A recent study reported larger ratios, finding about 40% of MD multi-unit activity increase with SWRs that were coupled to spindles (*Yang et al., 2019*). In this study, spindles were detected from EEG skull screws using criteria that included at least 0.5 s of duration; spindle-coupled SWRs were then classified as those SWRs that occurred between the onset and offset of spindles. Our LFP recordings in mPFC show that time windows longer than 0.5 s are likely to encompass one or more KCs, which are correlated with SWRs and often followed by a rebound in thalamic firing. In rodents, spindles defined by a duration criterion that overlaps with the delta oscillation are likely to reflect KC correlations (in addition to thalamo-SWRs correlations), and could increase the fraction of thalamic electrodes positively modulated with SWRs. The direct recording of LFP and KCs together with the analysis of shorter, within-spindle, timescales helps disambiguate the multi-oscillation correlations.

Computationally, higher order thalamic nuclei (which include those in the midline [*Guillery and Sherman, 2002*; *Varela, 2014*]) have been proposed to gate communication between neocortical areas. This 'gate' may be implemented through direct control of cortical synchronization and effective connectivity (*Saalmann and Kastner, 2009*; *Saalmann, 2014*; *Jaramillo et al., 2019*), and through the regulation of the signal-to-noise ratio in resting states (*Yang et al., 2019*). The broad decrease in thalamic activity at the time of potential memory reactivation in CA1, could facilitate effective information transfer between hippocampus and neocortex by reducing interference from the main input (thalamic) to neocortical areas. The signal-to-noise ratio may be particularly favorable to the transmission of novel traces between CA1-mPFC during spindles, due to the enhanced participation of recently active CA1 cells in SWRs. The spindle oscillation may provide a protected, phase-organized (*Wilson et al., 2015*), functional window for select memory consolidation computations, for example those required to extract statistical regularities and promote synaptic plasticity. We find that there is an increase in thalamic bursting near SWRs, a firing mode that is more effective at activating post-synaptic cortical cells (*Swadlow and Gusev, 2001*). The function of thalamic bursting is not fully understood, but this thalamic response mode promotes better detectability of stimuli during wakefulness (*Lesica, 2004*; *Alitto et al., 2005*; *Ortuño et al., 2014*), and has been proposed to serve as a detection signal that can facilitate plasticity and learning in thalamocortical networks (*Crick, 1984*; *Varela and Ahmad, 2019a*).

## Functional windows for multi-region coordination in non-REM sleep

The up-states of the non-REM slow and delta oscillations are thought to provide discrete windows of hundreds of milliseconds in which hippocampal-neocortical interactions can occur (*Destexhe et al., 2007*; *Penagos et al., 2017*). Hippocampal SWRs happen mainly during up-states and nested within the cycles of spindles, and spindles are themselves largest in amplitude following the transition from the down-to-up state (*Peyrache et al., 2011*; *Gonzalez et al., 2018*). Thalamic cells often showed a rebound in firing after mPFC KCs (LFP markers of the slow oscillation trough), suggesting that SWRs that occur during the down-to-up state transition overlap in time with an increase in thalamic drive. The thalamic cells with post-KC rebound could influence CA1-mPFC interactions by influencing the dynamics within the slow oscillation up-state. Several reports have shown that subsets of neocortical and TRN cells are preferentially active at the start or at the end of neocortical slow oscillation up-states, although it is unclear what drives this differential activation (*Gardner et al., 2013*; *Puig et al., 2008*; *Fanselow and Connors, 2010*; *Batterink et al., 2016*). Our data raise the possibility that cells in the dorsal thalamus (which can drive neocortical and TRN cells) could provide excitation early on in the slow oscillation up-state and influence the ensuing dynamics (*Neske, 2015*). Cells in the thalamic nuclei that we recorded from project densely to superficial layers of the neocortex, where they can influence deeper layers through their synapses on the dendritic tufts of pyramidal cells or via layer I interneurons (*Rubio-Garrido et al., 2009*; *Schuman et al., 2019*).

At finer time scales, individual SWRs events have durations that are strikingly similar to the period of the spindle oscillation (close to 100 ms), within the temporal range of spike-timing-dependent

synaptic plasticity (*Buchanan and Mellor, 2010*; *Sadowski et al., 2016*). Indeed, SWR- or spindle-like stimulation patterns induce LTP respectively in CA3-CA1 (*Sadowski et al., 2016*) and in neocortical synapses (*Rosanova and Ulrich, 2005*). These cell ensemble and circuit level oscillations and the nesting of SWRs in the spindle cycles can therefore induce the long-term synaptic remodeling behind memory consolidation. Both synaptic enhancement and weakening have been suggested to occur in non-REM sleep and contribute to consolidation (*Frey and Morris, 1997*; *Redondo and Morris, 2011*; *Puentes-Mestril et al., 2019*; *Seibt and Frank, 2019*; *Tononi and Cirelli, 2020*; *Langille, 2019*). Although our preparation does not address the synaptic level, our results demonstrate that CA1 cells that were active during exploratory behavior have enhanced association with SWRs and are phase-locked to the mPFC spindle oscillation. An enhanced activation of task-specific cells during SWR-spindles, in temporal windows that can promote synaptic plasticity, may explain the increase in memory following spindle-like stimulation (*Latchoumane et al., 2017*), and suggests that spindle oscillations provide windows for synaptic reorganization of recently acquired memories. We observed an increase in spindle-CA1 correlations after behavioral experience even in the absence of significant changes of spindle-thalamic correlations. It is possible that behavioral tasks with stronger memory demands would reveal a modulation of thalamic cells in addition to the observed CA1 modulation. Still, the differential enhancement of CA1-mPFC relative to thalamo-mPFC correlations could contribute to differential synaptic plasticity changes in these two pathways (*Puentes-Mestril et al., 2019*; *Tononi and Cirelli, 2020*). Spindles could provide a thalamocortical phase-organizing mechanism for communication between CA1 and mPFC that could lead to modification of CA1-mPFC synapses following behavioral tagging. Similar phase-specific communication mechanisms have been proposed during awake behavior, in which the phase of the theta rhythm may help organize long-range and pathway specific exchange between hippocampus and cortex (*Sirota et al., 2008*; *Colgin et al., 2009*; *Hasselmo, 2005*).

The results suggest that the dynamics of thalamic cell activation in non-REM results from a complex interaction between spindles, SWRs, KCs and previous experience. Future studies could incorporate these covariates to model thalamic firing and provide insight regarding the interactions between these factors and their role across the population of thalamic cells and nuclei (*Truccolo et al., 2005*; *Chen et al., 2011*).

## Sparse firing of thalamic cells during spindles and its role in systems memory consolidation

Sparse firing of thalamocortical cells during spindles has been reported before (*Huguenard and McCormick, 2007*; *Huguenard and McCormick, 2007*; *Steriade and Deschenes, 1984*; *von Krosigk et al., 1993*; *Crunelli et al., 2018*; *Steriade, 1994*), but it was unclear how multiple cells in the dorsal thalamus would correlate with each other, and its relevance for the systems consolidation of memory has not been discussed. We find that pairs of thalamic cells are consistently phase-locked to the mPFC LFP within the period of spindle oscillations, suggesting that individual spindle cycles group and organize the activation of different thalamocortical ensembles at consistent phase lags. The repeated activation of thalamic cell ensembles through non-REM sleep could promote spike-timing-dependent synaptic plasticity. An intriguing hypothesis is that the sparse firing at the single-cell level and phase-locked firing of groups of cells could lead to the interleaved excitation of different sets of thalamocortical synapses in subsequent spindle cycles, and in coordination with the reactivation of recently active CA1 place cells. The activation of varying combinations of thalamocortical synapses with CA1 SWRs that engage recently active cells (*Figure 9*) may promote a broad range of synaptic associations at the neocortical level, make the neocortical trace more robust to noise, and facilitate generalization. Critically, the sparse activation could prevent plasticity saturation (*Watt and Desai, 2010*) and contribute to gradual synaptic changes that would allow for extended computations (*Penagos et al., 2017*) and facilitate continuous learning. One of the challenges in the development of artificial neural networks, has been to overcome the catastrophic interference that can occur during sequential training. While real-world settings require continuous learning, artificial network training algorithms are not yet optimized for these conditions. Sparsity and controlled weight changes have been suggested to facilitate robust representations and learning (*Brunel and Hakim, 1999*; *Brunel and Hakim, 2008*; *Kirkpatrick et al., 2017*; *Hawkins and Ahmad, 2016*). Future studies of thalamocortical sleep dynamics can provide clues on the biological

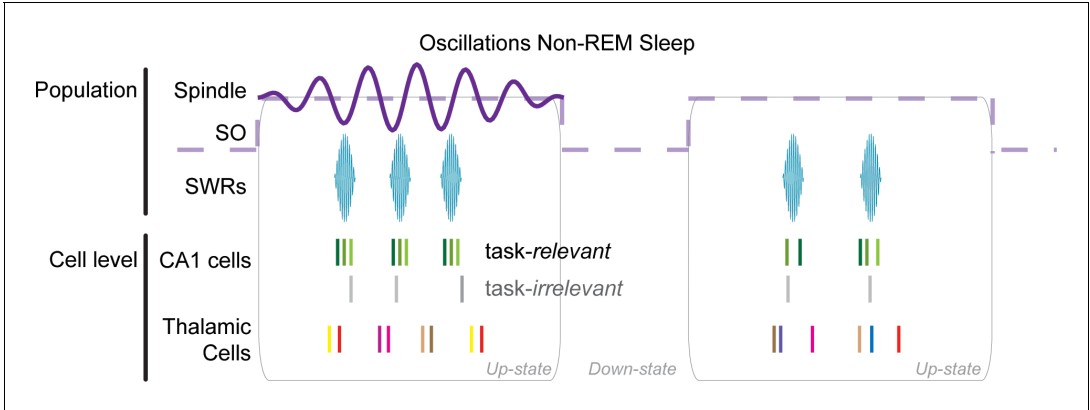

**Figure 9.** Model of spindle organization of thalamic and CA1 spike dynamics. Task-relevant CA1 cells are preferentially engaged in SWRs and phase-locked to the spindle oscillation, coinciding with sparse co-activation of thalamic cells. The activation of different combinations of thalamic cells in subsequent spindle cycles could favor a broader range of synaptic associations at the neocortical level, while the sparsity of firing could ensure incremental synaptic changes and promote continuous learning. Each color in the spike rasters represents spikes from a different cell. SO = slow oscillation.

implementation of continuous learning algorithms and help improve artificial networks and their applications.

Behaviorally, the systems consolidation of episodic memory leads to the stabilization of memories and to the formation of mental models or schemas that have higher adaptive value for the organism than the original details of the experience (*McClelland et al., 1995*; *Kumaran et al., 2016*). The oscillatory interactions between hippocampus and neocortex during sleep support the extraction of statistical consistencies and integration of hippocampal memory traces into gist-like neocortical schemas that are applicable beyond the specific conditions in which a memory was acquired (*Moscovitch et al., 2016*; *Penagos et al., 2017*). Sparse thalamocortical population synchronicity during spindles may therefore promote incremental synaptic changes that optimize the interleaved extraction and incorporation of implicit information from recently acquired hippocampal memory traces into the neocortical representations.

Among the nuclei in the limbic thalamus, most of the evidence regarding their role on memory comes from studies of the RE nucleus. Pharmacological and optogenetic inactivation impaired performance in delayed spatial alternation memory tasks (*Hallock et al., 2016*; *Hallock et al., 2013*; *Maisson et al., 2018*; *Viena et al., 2018*). RE contributes to working memory by coordinating activity in CA1-mPFC, and through encoding trajectory specific and head direction information during spatial navigation (*Ito et al., 2015*; *Jankowski et al., 2014*). In addition, the mPFC modulates the firing mode of RE cells (*Zimmerman and Grace, 2018*) and can use this thalamic link to help retrieve memory traces (*Jayachandran et al., 2019*; *Kupferschmidt and Gordon, 2018*). Likewise, pharmacological blockade points to the role of RE in long-term memory (*Loureiro et al., 2012*). Although it has no direct projections to CA1, the MD nucleus also contributes to working memory possibly through its direct projections to mPFC (*Bolkan et al., 2017*). These studies provide key evidence to understand the role of higher order thalamic regions in active cognitive behavior. Recordings in RE suggest that, similar to its coordinating role in wakefulness, it may contribute to coupling oscillatory activity between mPFC and CA1 (*Ferraris et al., 2018*; *Hauer et al., 2019*). Our results extend these findings and point to spindle-specific modulation of cells in RE and other limbic nuclei of the thalamus as a function of CA1 ripple activity. The observed modulations in firing in higher order thalamic nuclei could enhance synaptic plasticity and memory. In addition, a few studies point to the role of RE in extinction (*Ramanathan and Maren, 2019*; *Xu and Südhof, 2013*); more generally, sleep can both increase and decrease synaptic strength (and recent evidence points to a role of sleep oscillations in forgetting [*Langille, 2019*; *Kim et al., 2019*]). Dynamic engagement in spindle oscillations could control which cell populations undergo up- or downregulation of synaptic strength.

In certain states, such as anesthesia or epilepsy (*McCormick and Huguenard, 1992*), cells in the thalamus may fire at the spindle population rhythm, and sparse synchronicity in the thalamus might

be a distinguishing feature of normal resting states (*Akeju et al., 2018*). The disruption of thalamocortical oscillations has been proposed as a pathophysiological model for several neuropsychiatric disorders (*Llinás et al., 1999*; *Fogerson and Huguenard, 2016*). Oscillatory stimulation (mimicking the abnormally increased delta rhythm in schizophrenia) of thalamic projections to CA1 disrupts working memory in rats (*Duan et al., 2015*), and spindle irregularities (reduced or abnormally enhanced spindles) have been observed in children with neurodevelopmental disorders (*Gruber and Wise, 2016*), in schizophrenia (*Wamsley et al., 2012*), and in absence epilepsy (*Lüttjohann and Pape, 2019*). A careful balance in the excitatory and inhibitory thalamic networks is required to generate the appropriate population level synchronicity and function. Studying spindle dynamics at the cellular and population level will thus be critical to understand the pathophysiology behind these disorders, and to design therapeutic interventions that preserve normal synchronicity in the thalamus as well as the multi-region coordination that underlies cognitive function (*Manoach et al., 2020*).

## Conclusion

Memory consolidation promotes the stabilization and incorporation of memory traces initially stored in the hippocampus into neocortical networks, contributing to the development and training of neocortical models that allow adaptive behavior. Recording from naturally sleeping rats, we present results that suggest that the individual cycles of the spindle oscillation provide a key processing time window that combines and organizes the activation of recently active CA1 cell ensembles, as well as subsets of cells in thalamocortical networks. The rebound activation of thalamic cells following down-states could facilitate up-state dynamics. As the cycle of the slow oscillation progresses, the enhancement of spindle-CA1 activity relative to spindle-thalamic activity following spatial experience could bias transmission in CA1-mPFC compared to thalamo-mPFC pathways. Lastly, the sparse and interleaved activation of groups of thalamic cells during spindles may provide a critical mechanism to facilitate the continuous integration of novel hippocampal memories into neocortical schemas for generalization and lifelong learning.

# Materials and methods

We used eight male Long-Evans rats (400–500 g) in the experiments reported here (13 additional animals were implanted with multi-electrode arrays as part of the same project, but did not fulfil the criteria for inclusion in the current report due to lack of clear histological identification of electrode location in thalamus, or lack of electrophysiological signals in all three brain regions under study). Animals were housed individually, provided with food and water ad libitum, and monitored in a temperature-controlled room with a 12 hr light/dark cycle (lights on/off at 7:00 am/7:00 pm). These experiments were approved by the Committee on Animal Care at the Massachusetts Institute of Technology and conform to US National Institutes of Health guidelines for the care and use of laboratory animals.

## Electrode arrays, surgical implantation and electrophysiology recordings

Multielectrode arrays (microdrives) with up to 21 nichrome wire tetrodes, 18 tetrodes for recording and three references, one per brain region (individual tetrode wires were 12.5 μm in diameter). The microdrives were prepared according to standard procedures in the laboratory, and the base was modified from previous designs (*Kloosterman et al., 2009*; *Nguyen et al., 2009*) to target the three regions of interest. A diagram of a typical electrode configuration is shown in *Figure 1a*. The tetrodes spanned the following coordinates from Bregma (*Paxinos and Watson, 1986*): Anterior 3.65 mm to 2.4 mm for mPFC, posterior 1.4 to 3.2 mm for the thalamus, and P3.1 to 4.4 mm for dorsal CA1. The arrays for mPFC and thalamus were positioned at 16 degrees from the vertical axis to prevent damage to the sagittal sinus. Sterile surgery procedures were performed for chronic microdrive implantation; anesthesia was induced by an intraperitoneal injection of a solution of ketamine (25 mg/kg), xylazine (3 mg/kg) and atropine (0.027 mg/kg), and maintained with 1–2% inhaled isoflurane. Up to eight bone screws were secured to the skull for support; three craniotomies were drilled over the target coordinates and the dura mater membrane was removed to allow for tetrode penetration. During surgery, the most anterior thalamic electrode was placed above P1.4 mm and lateral 1.8 mm, and the linear thalamic array was aligned parallel to the sagittal suture; this positioned the

rest of the electrodes over their corresponding craniotomies. The implant was secured to the skull with dental cement after surrounding the exposed craniotomies and tetrodes with silicon grease to keep them from being fixed by the cement. Animals were preemptively injected with analgesics (Buprenorphine, 0.5–1 mg/kg, subcutaneous), and monitored by the experimenter and veterinarian staff for 3 days post-surgery.

The tetrodes were gradually advanced to the target depths over the course of 1.5–2 weeks before recordings started; the slow advancement of electrodes facilitated stable unit and local field potential (LFP) recordings over several hours. Once the target depth was reached, recordings were performed if units were present, and electrodes were lowered further in non-recording sessions to search for new units and allow for electrode drift before the next recording session. Recording sessions typically lasted at least 3 hr, during which the animal was in a quiet squared enclosure of about 50 × 50 cms, were they cycled through several bouts of sleep and wakefulness. Spikes were acquired at 32 kHz after the signal from any of the tetrode channels crossed a preset threshold. Individual units were isolated by manual clustering on peak spike waveform amplitudes across all channels using custom spike sorting software (Xclust; M.A.W.; available at https://github.com/wilsonlab/mwsoft64/tree/master/src/xclust) (*Xclust, 2013*). LFPs were sampled at 600 Hz after anti-aliasing filtering and downsampling from up to 3.125 kHz. Two infrared LEDs (sampled at 30 Hz) attached to the headstages of the acquisition system (Neuralynx) were used to track the rat's position and estimate velocity. Part of this data set has been made available through the Collaborative Research in Computational Neuroscience data sharing website (*Varela and Wilson, 2019b*).

## Spatial exploration in the radial maze

In preparation for the six behavioral sessions, three of the implanted rats were food restricted for at least 5 days before behavior started. In each session, rats explored a radial maze (with four arms for one rat and eight arms for the other two rats) that was surrounded by consistent visual cues on the walls of the room; rats retrieved one 45 mg sucrose pellet every time the animal reached the end of the arm (provided by the experimenter). The behavioral exploration (two sessions per animal) lasted 30 min, and the recordings included at least 30 min of sleep pre- and post-behavior. All the rats had less than 7 days of experience exploring the maze at the time of these recording sessions. For one of the rats, the two behavioral sessions were days 1 and 2 of exploration (no previous maze experience before day 1), for rat 2, they were days 2 and 4, and for the last rat they were days 3 and 6. Therefore, all the recordings were performed during early experience in the mazes, when memory traces are expected to be dependent on the hippocampus for retrieval (*Varela et al., 2016*). We did not observe differences between the first and second session in each rat, and the data were pooled together.

## Selection of electrodes and sessions for analyses

Sessions were selected based on having at least one unit in two of the three regions of interest, clear sleep events (KCs, spindles, SWRs) in the neocortical and hippocampal LFPs, and clear histological confirmation of thalamic tetrode location. While CA1 electrode location can be confirmed via electrophysiological LFP features alone, we find that LFP signals alone do not reliably allow specific localization within the nuclei of the dorsal thalamus and histological confirmation is necessary. For arrays with independently moving tetrodes, this requires the identification of electrode lesions for all the tetrodes in the array to prevent ambiguity when matching signals to electrode position. Thus, for these experiments we only include data from rats in which we successfully confirmed the histological location of all implanted thalamic (and mPFC) tetrodes, and CA1 location was confirmed electrophysiologically. The location of electrodes used for analyses is reported in *Figure 1—figure supplements 1–2*. To prevent repeated counts of the same units across sessions, unit analyses include only units for which 1) the tetrode had been moved from the previous recording session at least 80 μm or 2) the amplitude projections used for spike sorting had different profiles compared to the preceding recorded session.

## LFP state and event detection

For each rat, we selected one wire from one tetrode with clear sleep events (determined by visual inspection) for LFP analyses (one tetrode per brain region). To detect reference events for analyses,

the three LFPs were filtered (with zero-phase distortion) in the delta (1–4 Hz), spindle (6–14 Hz), and ripple (100–275 Hz) bands using a band-pass finite impulse response filter designed with a blackman window and filter order six times the sampling rate/bandwidth ratio.

We detected reference events during states in which the power of both the delta, in the mPFC LFP, and ripple bands, in CA1, were above the mean for the session for at least two minutes; animal velocity and behavior during these time periods confirmed that the animal was immobile during detected offline states (*Figure 1*). These electrophysiological and behavioral traits are defining features of non-REM sleep. The position information to calculate velocity was obtained from the LEDs attached to the headstages; gaps in the position information (due to occlusion of the LEDs) shorter than 300 ms were linearly interpolated using nearby values; gaps longer than 300 ms were not considered for further analysis. The space derivative was used to calculate velocity and then smoothed (Gaussian kernel, 250 ms standard deviation) and linearly interpolated to up-sample and match the 600 Hz of the LFP.

Non-REM periods detected with these combined criteria of high delta and SWR power, showed large transient deflections in the mPFC LFP, or K-complexes (KCs). To detect KCs, we first rectified the filtered (1–4 Hz) LFP (positive values were made equal to 0), the resulting trace was squared and KCs detected as local maxima, when the amplitude was above the mean plus four standard deviations (mean and standard deviation calculated from the LFP in quiet states). We detected spindle troughs as local minima, three standard deviations below the mean of the spindle-band filtered LFP (6–14 Hz). The SWR detection algorithm detected times when the squared, filtered LFP (100–275 Hz) had an amplitude above the mean plus three standard deviations for at least 20 ms (mean and standard deviation calculated for the LFP when the animal was quiet). If two SWRs were closer than 20 ms they were considered a single ripple event. The SWR timestamp was selected as the time with the largest absolute value in the ripple filtered LFP.

## Correlation quantifications: peri-event histograms and generalized linear model

We used peri-event histograms (PETHs) to quantify the temporal association between sleep events (trough of KCs, spindles, peak of ripples) and the unit activity in each brain region. To compare across sessions and animals, we quantified the change (in units of standard deviation) of the event-triggered average with respect to a null distribution calculated through a permutation test. Specifically, we first calculated the average peri-KC, peri-spindle and peri-SWR histogram across detected events in the recording session (bin 15 ms, followed by smoothing with a Gaussian kernel with 15 ms standard deviation). All the event timestamps were then shifted by a number between 1.5 and 2.5 s (randomly chosen with uniform probability) to offset the event times with respect to the local spikes, while preserving the inter-event time intervals. The average PETH (15 ms bins, Gaussian smoothed) was calculated from the shifted event timestamps, and this procedure was repeated 100 times to generate a distribution of expected PETHs under the null hypothesis of no correlation between sleep events and local activity (*Figure 1—figure supplement 4*).

To characterize the relation between thalamic firing and SWR rate more precisely, we implemented generalized linear models (GLM) of CA1 spikes and SWRs as a function of thalamic spikes. Multi-unit spikes from CA1 and the thalamus, and CA1 SWRs, were binned through the non-REM sleep periods in each session (bin 250 ms). A GLM was fit in matlab (*Equation 1*) to estimate the log of counts of spikes or SWRs in CA1 ($\mu$) as a linear function of the spikes in the thalamus (X), assuming a Poisson distribution. From the fitted model, we calculated the estimated change in the response variable (firing in CA1) per unit change of the predictor (firing in thalamus) from the equivalent *Equation (2)*. The Wald test was used to test the significance of the thalamic parameter ($\beta_1$).

$$\log(\mu) = \beta_0 + \beta_1 X \tag{1}$$

$$\mu = \exp(\beta_0 + \beta_1 X) \tag{2}$$

## Phase-locking analyses

We used some of the phase analyses methods as in *Siapas et al., 2005*, and matlab's circular statistics toolbox for quantification and statistical testing. We estimated the instantaneous phase using the Hilbert transform of the mPFC LFP filtered in the delta and spindle bands. Because asymmetry in

the LFP shape can bias the phase distributions, the distribution of phases at the time of spikes or SWRs was normalized by the distribution of phases for the full delta or spindle-filtered LFP during non-REM sleep. After normalization, we used the Rayleigh test to determine if the angular phase at the spikes or LFP events timestamps were significantly different from a uniform distribution. A fit to the Von Misses distribution was then used to estimate the mean vector direction (mean preferred phase), and the degree of phase-locking from the concentration parameter *kappa*. The concentration parameter is a reciprocal measure of dispersion and increases as the density of values near the mean of the distribution increases. We also estimated phase based on linear interpolation between LFP peaks and troughs (arbitrarily assigned $-\pi$ and $\pi$ values; equivalent to the 'extrema' method in *Siapas et al., 2005*); we did not find differences in the phase-locking based on either method and the results are reported for phase estimates based on the Hilbert transform.

Other statistical comparisons were made using non-parametric tests, the Mann-Whitney U-test for unpaired data, and the Wilcoxon signed ranked test for paired data (reported in the text for each specific analysis). We used Kruskal-Wallis for one-way ANOVA and two-way ANOVA to test interactions between two variables and an F-test to assess equality of variances; for multiple pairwise comparisons, we used the Tukey-Kramer test. Whisker plots are used to display population distribution of statistics. The central mark indicates the median, and the bottom and top edges of the box indicate the 25th and 75th percentiles, respectively.

## Acknowledgements

We thank Jorge Jaramillo, David Theurel and Fabian Kloosterman for comments and feedback that helped improve an earlier version of the manuscript, and Scout Brisson for help preparing *Figure 1—figure supplements 1–2*. We also thank the funding sources: support was provided by the Caja Madrid Foundation (Postdoctoral Fellowship to C.V.), and by the Brain and Behavior Research Foundation (NARSAD Young Investigator Award to C.V). This work was also supported by the Center for Brains, Minds and Machines (CBMM), funded by NSF STC award CCF-1231216, and by NIH grant TR01-GM10498.

## Additional information

### Funding

| Funder | Grant reference number | Author |
| --- | --- | --- |
| Caja Madrid Foundation | Convocatoria 2008 | Carmen Varela |
| Brain and Behavior Research Foundation | 22852 | Carmen Varela |
| National Science Foundation | STC award CCF-1231216 | Matthew A Wilson |
| National Institutes of Health | TR01-GM10498 | Matthew A Wilson |

The funders had no role in study design, data collection and interpretation, or the decision to submit the work for publication.

### Author contributions

Carmen Varela, Conceptualization, Data curation, Software, Formal analysis, Funding acquisition, Investigation, Methodology, Project administration; Matthew A Wilson, Conceptualization, Supervision, Funding acquisition, Investigation, Project administration

### Author ORCIDs

Carmen Varela ![ORCID] https://orcid.org/0000-0003-0398-2567

### Decision letter and Author response

Decision letter https://doi.org/10.7554/eLife.48881.sa1
Author response https://doi.org/10.7554/eLife.48881.sa2

## Additional files

### Supplementary files
• Transparent reporting form

### Data availability
Data files are available through the CRCNS website ('HC-24'; https://doi.org/10.6080/K0K35RVG); this includes data sets with raw data (LFP and units) recorded from 3 sessions, derived data (such as sleep-related events, like K-complexes, ripples and spindle cycles) as well as several matlab code to illustrate the main findings in the current manuscript. The data set and the documentation that describes it in detail is available through the CRCNS website ('HC-19' data set; http://crcns.org/data-sets/hc/hc-19).

The following dataset was generated:

| Author(s) | Year | Dataset title | Dataset URL | Database and Identifier |
|---|---|---|---|---|
| Varela C, Wilson MA | 2019 | Simultaneous extracellular recordings from midline thalamic nuclei, medial prefrontal cortex and CA1 from rats cycling through bouts of sleep and wakefulness | https://doi.org/10.6080/K0K35RVG | Collaborative Research in Computational Neuroscience, 10.6080/K0K35RVG |

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
