## [Decision Letter]

**Acceptance summary:**

This manuscript includes technically challenging simultaneous recordings from mPFC, hippocampus, and thalamus and provides novel insights about coordination of activity across these regions during sleep. The findings impact theories of memory consolidation operations during non-REM sleep. The findings are likely to be of interest to many learning and memory researchers as well as researchers who are studying the functions of sleep.

**Decision letter after peer review:**

Thank you for submitting your article "mPFC spindle cycles organize sparse thalamic activation and recently active CA1 cells during non-REM sleep" for consideration by *eLife*. Your article has been reviewed by three peer reviewers, and the evaluation has been overseen by Laura Colgin as the Senior and Reviewing Editor. The reviewers have opted to remain anonymous.

The reviewers have discussed the reviews with one another and the Reviewing Editor has drafted this decision to help you prepare a revised submission.

Summary:

This manuscript reports findings that shed light on the relationship between oscillations (i.e., thalamocortical spindles, hippocampal sharp-wave ripples, and neocortical slow oscillations) and single unit firing in the limbic thalamus, mPFC, and hippocampus. The manuscript includes two major observations; (1) Spikes of thalamic neurons are phase-locked to cortical delta and spindle oscillations and are also modulated by hippocampal sharp wave ripples (SWRs); and (2) CA1 units that were active in preceding behaviors are also more active during cortical spindles that are associated with high SWR power. Based on these observations, the authors conclude that cortical spindles provide the time window for organizing the activity of recently active CA1 ensembles as well as sparse thalamocortical ensembles. This organized activity is hypothesized to contribute to the consolidation of recently acquired memory traces in the neocortex. Although no evidence of memory consolidation is provided in the manuscript, the authors provide several lines of analysis to support the hypothesis. To obtain these results, the authors implanted up to 21 tetrodes in the three regions to record from the three regions simultaneously, which is technically challenging and impressive.

However, reviewers were unconvinced that the conclusions were well supported by the results as presently analyzed. Reviewers were particularly concerned about the low number of cells and lack of clarity about where cells were recorded. Still, after a consultation session, reviewers agreed that the authors could potentially deal with these concerns without collecting new data. It should be noted, however, that if some of the additional analyses fail to support the authors' conclusions, it may become necessary for the authors to acquire more data or to change the paper's conclusions. If this occurs, the authors may choose to withdraw the paper and resubmit at a later time, considering that *eLife* typically only allows a two month period for revisions.

Essential revisions:

1) Hippocampal-cortical interactions during non-REM sleep have been previously studied by multiple labs, including the senior author's lab. Therefore, the most novel part of this study was viewed as the inclusion of recordings from the nucleus reuniens. However, only 23 units were recorded from reuniens (24 more units were recorded in other thalamic regions, such as mediodorsal, ventromedial, centromedial, etc. which could have distinct functions in memory). Reviewers understand that it is highly challenging to record single units from these deep brain regions, but still they were unconvinced that such a small number of recordings is sufficient to support the conclusions as written (in particular, Figures 4, 6, 10). Reviewers also agreed that it is odd to analyze all limbic thalamus neurons together when it is known that the different nuclei have different projections, inputs, and behavioral correlates. The authors should emphasize this point more strongly in the manuscript and discuss how coupling to oscillations within this circuit relates to the individual contributions of each thalamic nucleus to memory. Moreover, the analysis on thalamic neuron pairs is based on a very small sample size (n = 12 pairs). Only 6 pairs displayed co-activation and, again, it is unclear from which thalamic nucleus/nuclei these neurons were recorded. Recordings from the same or different thalamic nuclei could lead to different interpretations of coordination within or across brain regions.

2) In a previous study by Peyrache et al., 2011, it was shown that mPFC units are less responsive to SWRs during spindles compared to SWRs outside spindles, and it is unclear how to reconcile these findings with the conclusion of this manuscript that SWR-coupled spindles act as an organizer of memory consolidation. As the authors recorded from neurons in mPFC, it was suggested to check how mPFC units respond to SWRs within and outside spindles and to thoroughly discuss how these results relate to this earlier study.

3) In Figure 6, the authors claimed sparse and consistent activation of thalamic spikes, because of lack of rhythmicity in individual thalamic units in spite of clear rhythmicity in MUA. However it is possible that this is simply due to low-firing rates of individual neurons (e.g. Barry et al., Nature, 2012). Furthermore, in Supplemental Figure 5, some neurons appear to exhibit rhythmic firing. As this result forms one of the main conclusions of the manuscript, the authors should test the idea more carefully. For example, one possibility would be to implement a quantitative analysis of the degree of rhythmic firing by using curve fitting method in autocorrelation plots for individual cells (e.g. Royer et al., 2010; Brandon et al., 2013). Also, to correct the influence of the spike number, the autocorrelation of MUA should be analyzed with a reduced number of spikes, so that the number of spikes in autocorrelation plots is the same between groups. Furthermore, it would be interesting to examine whether spike cross-correlations of pairs of neurons differ within and outside spindles, to assess if the co-activity patterns are dynamic or hardwired.

4) In Figure 8, the authors claimed that the degree of rate modulation of thalamic cells during spindles increased when associated SWRs had higher power. However, is there any relationship between SWR power and spindle power? A concern was raised that the observed effect of SWR power could be an indirect effect of associated spindle power modulation. For example, high-power spindles can be detected more easily at precise timing, and consequently, the rate modulation may be observed more clearly. Also, although an enhanced coupling between thalamic neurons and high-ripple-power spindles (compared to low-ripple-power spindles) was reported in Figure 8, there was no change in thalamic neuron-SWR coupling with or without the presence of spindles, in contrast with a recent finding (Yang, Logothetis and Eschenko, 2019). Does the activity of these thalamic neurons correlate with spindle power? It is not clear how these results can be interpreted in the existing framework of the CA1-mPFC-thalamus network. Similarly, in Figure 9C, the authors should carefully check the relationship between SWR power and spindle power, and in Figure 9D, should examine the change of SWR power within and outside of spindles, to exclude the possibility of indirect influences on the main conclusions.

5) In Figure 9D, Rayleigh's z were assessed between neurons with different firing rates. However, the Rayleigh test is sensitive to the number of spikes (e.g. Sirota et al., 2008), and thus the authors should confirm the main conclusions with subsampled spikes such that the numbers of spikes are the same between groups.

6) The authors claim that there's an "increase in the amplitude (peak-to-trough) and rate (per second) of SWRs during sleep after the maze exploration". Yet, appropriate statistics to support this claim were not presented. Reviewers were not convinced that there would be a significant increase in SWR amplitude or rate when compared to data from proper control experiments.

[Editors' note: further revisions were suggested prior to acceptance, as described below.]

Thank you for re-submitting your article "mPFC spindle cycles organize sparse thalamic activation and recently active CA1 cells during non-REM sleep" for consideration by *eLife*. Your article has been re-reviewed by three peer reviewers, and the evaluation has been overseen by Laura Colgin as the Senior and Reviewing Editor. The reviewers have opted to remain anonymous.

Although many of the concerns raised in the initial review have been addressed, some of the major issues raised by the reviewers were not adequately addressed yet. The remaining concerns are detailed below.

Essential revisions:

1) Regarding the reviewers' major concern on the small number of thalamic units, the authors provided only justification of the small unit number, but not additional support of the main conclusions. The reviewers' concern was whether the main conclusions can be supported by the authors' data with a limited number of units, because some of their claims are too strong based on the presented data. For example, in the Abstract, the authors mentioned that "Thalamic units…fired at consistent lags with other thalamic units". But from their data, convincing evidence of consistent time lags of thalamic firing in the cross-correlation analysis (only one or two examples may show a "lag" out of 12 pairs) is still lacking.

2) Based on the reviewers' concern on their conclusion of lack of rhythmicity in individual thalamic units, the authors provided additional analyses. However, first, the subsampling procedure performed in this revision is not satisfactory, as the spike number of MUA (12000 or 6700) is still far larger than the typical spike number of units (2000 or less?). The reviewers originally suggested to equalize the spike number between groups, which has not been performed. The authors actually confirmed that "a smaller sample of spikes can reveal variations", and the subsampling procedure is meant to assess the impact of such variation due to a small sample number on the statistical test (e.g. Figure 4—figure supplement 1F).

In addition, the new analysis (Figure 4—figure supplement 1F) indicates that many units indeed fire at the rhythm of spindle oscillations, while other units fire at longer intervals. I think that this result contradicts the authors' conclusion of "the lack of rhythmicity in individual thalamic units".

On the other hand, if there is an organized (grouped) sequential reactivation as the authors suggested, the cross-correlograms should reveal both co-activation and suppression at specific intervals of the spindle cycles, which did not appear in the results. Therefore, the results only suggest that thalamic neurons fire at the spindle rhythm, but not necessarily at every spindle cycle (similar to theta-cycle-skipping cells). However, these results alone do not support organized coactivation nor sequential activation as described in the authors' claims as follows: "different groups of thalamic units contribute to activity in different spindle cycles" (subsection Sparse thalamic firing and consistent activation of cell pairs during spindles”); "individual spindle cycles group and organize the sequential activation of thalamocortical ensembles" (subsection “Sparse firing of thalamic cells during spindles and its role in systems memory consolidation”); and "the sparse firing at the single-cell level will allow for the interleaved excitation of different sets of thalamocortical synapses in subsequent spindle cycles" (subsection “Sparse firing of thalamic cells during spindles and its role in systems memory consolidation”).

3) In the original review, the reviewers did not ask which factor, either the spindle or ripple power, contributes to the stronger modulation of thalamic activity, but the question was whether the grouping into high and low ripple power (as in the main figures) may have associated confound of spindle power differences (e.g. one group may have a higher spindle power than the other). The authors only provided the analyses in which one variable was handled independently, but they should take into account interactions between two variables, ripple and spindle power. For example, one can use a regression model of the thalamic firing modulation with both spindle and ripple power, to assess the contribution of each factor.

4) In the original review, the question was not whether the mean of phase distributions changed or not by subsampling, but rather that Rayleigh's z values should be compared with the same number of spikes between groups (e.g. in their new Figure 7D).

---

## [Author Response]

Essential revisions:1) Hippocampal-cortical interactions during non-REM sleep have been previously studied by multiple labs, including the senior author's lab. Therefore, the most novel part of this study was viewed as the inclusion of recordings from the nucleus reuniens. However, only 23 units were recorded from reuniens (24 more units were recorded in other thalamic regions, such as mediodorsal, ventromedial, centromedial, etc. which could have distinct functions in memory). Reviewers understand that it is highly challenging to record single units from these deep brain regions, but still they were unconvinced that such a small number of recordings is sufficient to support the conclusions as written (in particular, Figures 4, 6, 10). Reviewers also agreed that it is odd to analyze all limbic thalamus neurons together when it is known that the different nuclei have different projections, inputs, and behavioral correlates. The authors should emphasize this point more strongly in the manuscript and discuss how coupling to oscillations within this circuit relates to the individual contributions of each thalamic nucleus to memory. Moreover, the analysis on thalamic neuron pairs is based on a very small sample size (n = 12 pairs). Only 6 pairs displayed co-activation and, again, it is unclear from which thalamic nucleus/nuclei these neurons were recorded. Recordings from the same or different thalamic nuclei could lead to different interpretations of coordination within or across brain regions.

Figure 4—figure supplement 2 (pair cross-correlograms) now includes the thalamic nucleus of origin for the units. While the study of specific thalamic nuclei is important, given the challenges of recording units from small thalamic nuclei in behaving rodents and simultaneously with other brain regions, we have focused here on studying shared properties as they relate to non-REM sleep. All of the nuclei we recorded are higher order thalamic nuclei, bidirectionally connected with frontal cortical areas, regulated by similar sleep-dependent neuromodulatory systems, and they share circuit, physiological and molecular properties (Varela et al., 2014; Varela, 2014; Guillery and Sherman, 2001; Phillips et al., 2019). Likewise, although reuniens has the most well-studied functional correlations with hippocampus (Dolleman-van der Weel et al., 2019), the hippocampus projects to rostral parts of the thalamic reticular nucleus and to medial prefrontal cortex (Cavdar et al., 2008; Zikopoulos and Barbas, 2008; Swanson, 1981), which could indirectly influence nuclei other than reuniens. In other words, the limbic and higher order thalamic nuclei are functionally related to cortical and hippocampal regions and may share important principles that motivate their study as a group. Our results suggest that at least some of the thalamic correlates of non-REM sleep can be observed across limbic thalamic nuclei. Future experiments that investigate the contributions of specific nuclei can be more interpretable in the context of the data reported in this paper.

Regarding the yield of units in thalamus, several technical factors limit the single cell yield in this multi-site preparation; as mentioned by the reviewers, targeting small deep regions like the thalamus in conjunction with distant but functionally connected regions (mPFC, CA1) is challenging. We aimed to work only with data from rats in which the electrode final location in the brain, and the track traversed during the recordings were unambiguously identified in histological sections, reported in the raw pictures of brain sections, in the supplementary material (approximately 1 in every 3 implanted rats met this criterion and had sufficient single cell and LFP data during sleep to be included in the study). In all rats, we found that the amplitude of spikes from units in the thalamus was small compared to the background spike activity, and that the unit yield was much lower compared to cortex and hippocampus (often only 1 or 2 units could be recorded in a session in the tetrodes implanted in the thalamus). Another limitation is that, in order to gather at least one hour of natural sleep, it is sometimes necessary to record for several hours (even in the light phase), and units that were not stable through the duration of sleep epochs (evaluated during manual spike sorting) were not used for analysis.

We believe that the low yield of units does not diminish the data or results, as we aimed to record the highest number of quality units (given the experimental constraints above). There are other examples in the thalamic literature with similar yields (e.g., Gardner, Hughes and Jones, 2013). Although new high-density recording probes (e.g., neuropixels) increase the unit yield in different brain regions, experiments that target functionally related and anatomically distant structures are still a challenge for these new technologies. As the techniques to increase unit yield continue to improve, our results provide valuable guidance regarding the functions of a thalamo-mPFC-CA1 network that is receiving increasing attention.

2) In a previous study by Peyrache et al., 2011, it was shown that mPFC units are less responsive to SWRs during spindles compared to SWRs outside spindles, and it is unclear how to reconcile these findings with the conclusion of this manuscript that SWR-coupled spindles act as an organizer of memory consolidation. As the authors recorded from neurons in mPFC, it was suggested to check how mPFC units respond to SWRs within and outside spindles and to thoroughly discuss how these results relate to this earlier study.

We performed additional analyses on the units and multi-unit spike data recorded from mPFC. We calculated a peri-SWR histogram of mPFC unit activity with the same parameters used in Peyrache et al., 2011 (2 ms bins in a window of 400 ms before and after SWRs), and separated the peri-SWR histogram based on ripples that occurred within or outside spindle cycles and using the same sample size (same number of coupled and uncoupled SWRs). We z-scored the resulting histograms to the same shuffled distribution (after randomizing all SWR timestamps). The resulting histograms for the mPFC units are shown in Figure 6—figure supplement 3A; the left plot in the top row shows the peri-SWR histogram calculated for SWRs uncoupled to spindles (each row is one unit), and the right plot for coupled SWRs (rows in the same order as in panel A; color code is zscore). Following the quantifications in Peyrache et al., 2011, we calculated the mean z-score in the -40 to 40 ms around SWRs; we found that on average there was no significant difference for the population of mPFC units between SWRs coupled or uncoupled to spindles (z-score 0.17 ± 1.26 for coupled-SWRs and 0.24 ± 0.86; p = 0.46, Wilcoxon signed rank). Our sample of mPFC units contained units that increased and units that decreased their firing near SWRs (left side plot on Figure 6—figure supplement 3A-iv, includes the peri-SWR histograms of mPFC unit spikes for all SWRs using Peyrache et al., 2011 parameters). Interestingly, the units with the strongest correlation calculated with all SWRs, showed differential firing for SWRs coupled or uncoupled to spindles, such that their firing decreased with SWRs coupled to spindles (units with the strongest negative correlation also had the strongest modulation with SWRs uncoupled to spindles). Although these observations are based on a low number of units, it is possible that some mPFC units have the strongest firing rate modulation (up or down) during SWRs that are not coupled to spindles (as observed by Peyrache et al., 2011).

To assess the mPFC-SWR association at the multi-cell level, we repeated the peri-SWR histograms for multi-unit spikes (all spikes recorded by one tetrode). With this analysis we found stronger correlation of mPFC spikes with SWRs coupled to spindles (z-score 0.83 ± 1.34 compared to uncoupled, 0.09 ± 1.53; p = 0.002). We also found stronger correlations with CA1 firing (55.81 ± 28.40, 50.40 ± 30.51; p < 10^-3^; Wilcoxon signed rank), and no significant difference in the thalamic tetrodes (-64 ± 1.42, 0.34 ± 0.95; p = 0.17). Overall we find that, at the cell population level, both CA1 and mPFC increase their firing significantly during SWRs coupled to spindles, and that certain mPFC cells maybe more strongly modulated (up or down) during SWRs that are not coupled to spindles. This may reflect two different dynamics in mPFC, one driven by the spindle oscillation, which may engage cell population activity in coordination with CA1, and one, outside spindles, which is dominated by cell specific modulation during non-coupled SWRs. Experiments that record large populations of mPFC cells will provide the best design to test this hypothesis.

These results are summarized in the manuscript.

3) In Figure 6, the authors claimed sparse and consistent activation of thalamic spikes, because of lack of rhythmicity in individual thalamic units in spite of clear rhythmicity in MUA. However it is possible that this is simply due to low-firing rates of individual neurons (e.g. Barry et al., Nature, 2012). Furthermore, in Supplemental Figure 5, some neurons appear to exhibit rhythmic firing. As this result forms one of the main conclusions of the manuscript, the authors should test the idea more carefully. For example, one possibility would be to implement a quantitative analysis of the degree of rhythmic firing by using curve fitting method in autocorrelation plots for individual cells (e.g. Royer et al., 2010; Brandon et al., 2013). Also, to correct the influence of the spike number, the autocorrelation of MUA should be analyzed with a reduced number of spikes, so that the number of spikes in autocorrelation plots is the same between groups. Furthermore, it would be interesting to examine whether spike cross-correlations of pairs of neurons differ within and outside spindles, to assess if the co-activity patterns are dynamic or hardwired.

We have performed additional analyses to more thoroughly investigate the possibility of rhythmic firing in thalamic units and multi-unit spikes. To address the effect of different numbers of spikes on the autocorrelations, we re-calculated the autocorrelations of multi-unit activity (MUA) with a subset of the spikes (12000 spikes, based on the median number of unit spikes) and observed similar autocorrelations as when using all the spikes. In Figure 4—figure supplement 1 we plotted the multi-unit autocorrelations with all spikes (panel A; each color one session), and the average of autocorrelations calculated in blocks of 12,000 sleep spikes through the session (panel B), as well as the autocorrelations of units (panel C; each color one unit). Panels D and E display an example from one session, the autocorrelation of the MUA spikes is shown first for all spikes (D left histogram); the autocorrelations for each block of 12,000 spikes through the recording session are shown in panel Di, while Dii displays autocorrelations in blocks of 6700 spikes –the number of spikes produced during sleep by a thalamic unit recorded by the same electrode in the same session-. The autocorrelation of the thalamic unit is shown in E for all the spikes, and in blocks of 2,000 spikes (E, bottom plot). Although using a smaller sample of spikes can reveal variations in the strength of the autocorrelations in both the multi-unit and unit, the shape of the autocorrelation was not affected by sampling.

To quantify the autocorrelations, we tried fitting the autocorrelation curves first with models such as those used by Royer et al., 2010 and Brandon et al., 2013. These models used sine and cosine functions to capture the oscillatory activity of hippocampal neurons during hippocampal theta (they also used decaying exponentials to account for the time course of the autocorrelation at short and longer lags). These models were not a good fit to our thalamic data (an example is shown in Author response image 1, for one thalamic unit fitted using the same model as in Brandon et al., 2013), and we opted to use spline interpolation to smooth the autocorrelation curves before quantification (blue thick curves in the autocorrelation histograms of Figure 4—figure supplement 1). We calculated the time at the autocorrelation maximum (after the minimum that follows lag 0) and plotted the distribution for MUA and unit spikes (Figure 4—figure supplement 1F). Although not significantly different between units and MUA, we found that the peak of the MUA autocorrelation occurs on average at 88.00 ± 62.58 ms (91.60 ± 59.49 ms, when using a fraction of the spikes), whereas the interval is longer for the units, 160.65 ± 118.07 ms, suggesting faster firing for MUA compared to the units (p = 0.07, Kruskal-Wallis). We further inspected all unit autocorrelations and plotted the only single unit that had a strong indication of oscillatory activity (Figure 4—figure supplement 1H); the peak of the autocorrelation plot for this unit occurs at 140 ms, or ~7 Hz, on the lowest range of the spindle band. Figure 4—figure supplement 1G, shows the autocorrelation of units in which the maximum autocorrelation value falls within the lags that correspond to the spindle band (6-14 Hz); although these units (n = 7 or 22.6 % of the sample) have their strongest autocorrelation within the spindle band, their values were low (0.055 ± 0.013) and not different from other units (p = 0.31).

**Author response image 1. sa2fig1:** Example of cosine fit (as in Brandon et al., 2013) to the autocorrelation of a thalamic unit. X axis, time (s); y axis, correlation coefficient.

Regarding the specificity of the correlations between pairs of thalamic units, we calculated additional cross-correlograms selecting the unit spikes that occurred during periods of wakefulness as well as during sleep periods *between* detected spindles (where no spindles were detected for at least 5 s). These additional plots are included for all thalamic pairs in the modified Figure 4—figure supplement 2, and show that the strongest cross-correlations were observed during sleep spindles, suggesting spindle specific co-activation of thalamocortical networks.

In summary, this additional set of analyses suggest that few units in the dorsal thalamus contribute oscillatory activity to the spindle oscillation, and that this rhythm results from activity at the population level. In addition, the analysis of simultaneously recorded thalamic units suggests sleep-specific correlations between subsets of thalamic cells.

4) In Figure 8, the authors claimed that the degree of rate modulation of thalamic cells during spindles increased when associated SWRs had higher power. However, is there any relationship between SWR power and spindle power? A concern was raised that the observed effect of SWR power could be an indirect effect of associated spindle power modulation. For example, high-power spindles can be detected more easily at precise timing, and consequently, the rate modulation may be observed more clearly. Also, although an enhanced coupling between thalamic neurons and high-ripple-power spindles (compared to low-ripple-power spindles) was reported in Figure 8, there was no change in thalamic neuron-SWR coupling with or without the presence of spindles, in contrast with a recent finding (Yang, Logothetis and Eschenko 2019). Does the activity of these thalamic neurons correlate with spindle power? It is not clear how these results can be interpreted in the existing framework of the CA1-mPFC-thalamus network. Similarly, in Figure 9C, the authors should carefully check the relationship between SWR power and spindle power, and in Figure 9D, should examine the change of SWR power within and outside of spindles, to exclude the possibility of indirect influences on the main conclusions.

Spindle and ripple power are indeed associated (Author response image 2 shows the covariance of the ripple-filtered CA1 LFP and spindle-filtered prefrontal LFP in all sessions). To disentangle the modulation of thalamic activity due to increased ripple power and to increased spindle amplitude associated with ripple power, we compared the peri-spindle cycle modulation index of thalamic spikes for spindle cycles classified based on amplitude or on ripple power (above or below the median). The modulation index values were similar in spindles classified based on median amplitude or ripple power (p > 0.3, Wilcoxon signed rank), but increased significantly from low to high amplitude spindles (p = 0.005) compared to low to high ripple power spindles (p = 0.25), suggesting a stronger modulation by spindle amplitude compared to ripple power, as expected from the role of thalamic cells in spindle generation (modulation index distribution for the population in Figure 6D). If there is a functional relation with spindle amplitude or ripple power, spindles in the upper quartiles should lead to stronger modulation. We then calculated the fraction of thalamic units in which the increase in modulation in spindles within the upper quartile was higher than the increase observed with the median classification. The increase in modulation index was higher in the upper quartile of spindle amplitude in 29 % of thalamic units and with ripple power in 22.6 % units. The modulation values were higher for spindle amplitude compared to ripple power and positively correlated (average of 1.16 ± 0.57 in the upper quartile of spindle amplitude compared to 0.60 ± 0.26 in the upper quartile of ripple power; p = 0.04, Mann-Whitney); units with stronger modulation with spindle power had higher values of ripple modulation (Spearman coefficient > 0.7; p < 0.04). For spindle amplitude, the lower modulation values also became lower with the quartile classification, leading to a wider spread of the distribution (when spindles were classified based on quartiles compared to median; p = 0.04; F-test) which was not the case with ripple power (p = 0.36; F-test; Figure 6E). These results suggest that the strongest firing modulations in thalamic cells are related to the spindle oscillation, and that ripples in CA1 are associated with smaller changes in firing modulation in about 20 % of thalamic cells.

**Author response image 2. sa2fig2:** Cross covariance of spindle and ripple envelope for all sessions.

To find out if there are synergistic effects between spindle amplitude and ripple power, we then detected the spindle cycles with the highest amplitude (upper quartile of the distribution) and compared the modulation of thalamic cells when ripple power was high or low (above or below median). We run a complementary analysis in which we detected spindle cycles with the highest ripple power (upper 25 %) and compared the modulation of thalamic cells in cycles with low or high amplitude. For the population of cells we did not find evidence of a significant effect of ripple power on the modulation value of thalamic firing in high amplitude spindles (p = 0.94; Wilcoxon signed rank), while higher spindle amplitude led to higher modulation values on high ripple power spindles (p = 0.04). However, although the modulation values were not significantly different, a majority (61.29 %) of units showed a further increase in modulation index with ripple power (average increase 18.15 ± 21.94 %); instead, 32.26 % of the units increased their modulation with spindle amplitude (in the spindles with highest ripple power; average increase 30.00 ± 20.97 %). The percent change in the modulation index with ripple power and spindle amplitude were correlated (Spearman correlation 0.51; p = 0.003).

Altogether, the data is consistent with the hypothesis that spindle-related firing in thalamic cells is regulated in conjunction with changes in ripple activity in the hippocampus, and this process may dynamically engage specific subsets of cells in the midline thalamus. Figure 6 (previous Figure 8) has been modified extensively to incorporate this extension of the results. Figure 6F illustrates an increased modulation with ripple power in the same thalamic unit from Figure 6A-C. The peri-spindle histogram for spindle cycles in the upper quartile of the distribution of peak-to-trough amplitudes is split between those that have low or high ripple power; the two groups of spindles are not significantly different in amplitude (p = 0.45), yet the unit increases its firing modulation with highripple-power spindles (top plots); instead, the same unit shows reduced modulation with spindle amplitude for spindles with the highest ripple power (bottom). Figure 6—figure supplement 2 includes additional examples of units that increase and decrease their spindle modulation with ripple power and spindle amplitude. Although we observed clear modulations of firing with spindles, the absolute number of spikes for single units did not change significantly with spindles low/high in ripple power and neither with SWRs coupled/uncoupled to spindles (Figure 6G).

Regarding the Yang et al., 2019 study, the main difference between our results and those of Yang et al., 2019 is that they found a large proportion of neural activity in the MD nucleus positively correlated with spindle-coupled SWRs. In our sample, we found that 7 units (14.89 % of the 47 that were analyzed with SWRs) showed a pure increase in firing in the 200ms window around CA1 SWRs; this means an increase in firing larger than 2 standard deviations from the null distribution and that was not associated with a post-inhibition rebound. Another 4 units (8.51 %) had a large rebound (larger than 2sd) that followed soon after a strong decrease in firing (below -2sd) (an example is shown in Figure 3B); 3 of these units with rebound were recorded in the reuniens nucleus and 1 in the ventromedial. Unfortunately, we only have 4 units recorded in MD and we cannot directly compare our results to those of Yang et al., 2019; however, our results extend theirs to suggest that some thalamic units increase their firing rate with SWRs, and this often takes the form of a rebound from decreased firing. Although we have not seen specific increases in absolute firing rate with spindle-coupled SWRs, we have seen that some cells increase their firing modulation (up and downregulation) with high-ripple power spindles; Yang et al., 2019 describe relying on multi-unit neural activity for some of their analyses, and it is possible that their MUA recordings in MD reflect pooled increased modulation in certain cell populations. Our results suggest that individual cells may not display and absolute increase in firing with spindle-coupled spindles, instead, when hippocampus ripple activity is highest, the firing of some thalamic cells will both increase and decrease following the dynamics of the ongoing spindle cycles, suggesting phasespecific effects in thalamocortical synapses that could serve a function in facilitating the consolidation of hippocampus to neocortical traces.

**Author response image 3. sa2fig3:** Example of peri-event histogram smoothing based on moving average (red, “new” curve), and on Gaussian smoothing (blue, “old”), used prior to calculating unit modulation index.

It is also worth noting a difference with the spindle detection in our study; in the Yang et al. study, spindles were detected from EEG skull screws using criteria that included at least 0.5s of duration; spindle-coupled SWRs were then classified as those SWRs that occurred between the onset and offset of spindles. Our LFP recordings in mPFC show that time windows longer than 0.5s can encompass one or more KCs, which are correlated with SWRs and spindles, and can be followed by a thalamic rebound of firing. Spindles defined by this duration criterion could partly reflect KC-thalamic firing and increase the fraction of thalamic electrodes positively modulated with SWRs.

5) In Figure 9D, Rayleigh's z were assessed between neurons with different firing rates. However, the Rayleigh test is sensitive to the number of spikes (e.g. Sirota et al., 2008), and thus the authors should confirm the main conclusions with subsampled spikes such that the numbers of spikes are the same between groups.

We repeated the phase locking analysis of thalamic units to the spindle and delta filtered mPFC LFP with a random subset of 10 % of the spikes. We find that the circular mean of the newly calculated phase distributions does not change significantly (p = 0.68 for spindle and 0.07 for delta); the phase distributions with less spikes do have significantly different Z-values and modulation values (except kappa for delta, which is not significantly different). The distributions look qualitatively similar (Author response image 4).

**Author response image 4. sa2fig4:** Examples of phase distributions for two thalamic units. Distributions were calculated with 10% of the spikes (left plots) or with all spikes in sleep, as done for the results reported in the manuscript (right plots).

6) The authors claim that there's an "increase in the amplitude (peak-to-trough) and rate (per second) of SWRs during sleep after the maze exploration". Yet, appropriate statistics to support this claim were not presented. Reviewers were not convinced that there would be a significant increase in SWR amplitude or rate when compared to data from proper control experiments.

We have re-examined this result with a larger dataset and added control analyses. We had previously used 6 sessions from 3 rats to address the question of potential changes in SWR power and rate before and after spatial exploration. These recording sessions had been selected based on the presence of stable units in CA1 during run (and in the before and after run sleep recordings). To these recordings, we now added two more sessions from a different rat and two additional sessions from two of the previous rats, for a total of 10 sessions from 4 rats. We still find that the peak-totrough amplitude and the rate per second of detected SWRs is larger after run compared to before run (p = 0.027 and p = 0.019; Mann-Whitney test), even though the recording before and after run did not differ in total sleep time (p = 0.076). As a control for the possibility that these features change over time, we used another set of 4 longer recording sessions from two of the rats, in which we had recorded continuously (no maze exploration) for a similar overall amount of time. We compared the SWRs amplitude and rate in the first half to the second half of these sessions, finding no difference in amplitude nor rate between the first and second half of the recording (p > 0.2); the overall amount of sleep time was also not different in the first and second half of the recording, and not different from the pre-behavior or post-behavior sleep epochs in the other sessions with “run” epochs (p > 0.1 in all cases).This more thorough examination points to an enhancement of CA1 SWR activity immediately after spatial behavior, which may be related to the reactivation of the recently acquired episodic memory traces during rest.

**Author response image 5. respfig5:** 

[Editors' note: further revisions were suggested prior to acceptance, as described below.]

Essential revisions:1) Regarding the reviewers' major concern on the small number of thalamic units, the authors provided only justification of the small unit number, but not additional support of the main conclusions. The reviewers' concern was whether the main conclusions can be supported by the authors' data with a limited number of units, because some of their claims are too strong based on the presented data. For example, in the Abstract, the authors mentioned that "Thalamic units…fired at consistent lags with other thalamic units". But from their data, convincing evidence of consistent time lags of thalamic firing in the cross-correlation analysis (only one or two examples may show a "lag" out of 12 pairs) is still lacking.

We have increased the sample size for the pair analyses by adding 16 new units, for a total of 39 pairs (3.25 times more than in the previous version of the manuscript). These additional units had not initially been identified during manual clustering, which aimed to sort units in the three brain regions; the added units were detected (in 5 additional sessions from 4 of the rats) after revising the raw data looking for multiple units recorded simultaneously within the thalamus. The additional sessions allowed us to review pair correlations in a larger population of thalamic cells. The results and conclusions are still supported by the additional data. 33 % of the pairs had a peak above 3 standard deviations from the average cross-correlogram (average peak z-score 2.40 ± 1.21 sd, n = 39 pairs). We repeated the analyses of firing consistency between pairs with the added data, we calculated the smallest difference between the phase of the spikes in spindle cycles (Figure 1). The standard deviation of the distribution of phase lags was significantly lower for units with high spike correlations during sleep (pair cross-correlograms > 3 sd) compared to the units that were not temporally correlated. The standard deviation was significantly lower for correlated pairs compared to uncorrelated pairs (2.15 compared to 2.50; p < 0.001, Mann-Whitney U-test). These results have been updated in the text and in Figure 4—figure supplement 2 and Figure 5—figure supplement 1; the table with total unit numbers has been updated. We also updated the analysis of cross-correlation of unit pairs in low and high ripple power spindles. With the increased number of pairs, we found that the modulation of thalamic spindle firing with SWR power was stronger in half of the unit pairs; we calculated the modulation index for the cross-correlograms of thalamic pairs and found that it increased by an average of 26.64 % in 48.72% of the pairs in spindle cycles with high ripple power and decreased by 20.90 % in the rest of the pairs (n = 39).

2) Based on the reviewers' concern on their conclusion of lack of rhythmicity in individual thalamic units, the authors provided additional analyses. However, first, the subsampling procedure performed in this revision is not satisfactory, as the spike number of MUA (12000 or 6700) is still far larger than the typical spike number of units (2000 or less?). The reviewers originally suggested to equalize the spike number between groups, which has not been performed. The authors actually confirmed that "a smaller sample of spikes can reveal variations", and the subsampling procedure is meant to assess the impact of such variation due to a small sample number on the statistical test (e.g. Figure 4—figure supplement 1F).

We repeated the calculation of the autocorrelations with the subsampling procedure suggested by the reviewers: we equalized the MUA spike numbers to the number of spikes fired by the thalamic unit with the least number of spikes recorded in the same session. The autocorrelations of MUA spikes calculated with subsampling to the sparsest unit in each session still show a peak in the lags of the spindle frequency, and the maximum of the autocorrelation function (after removing lag 0) was on average at 119.3 ms for MUA, significantly lower than for units (average 223.6 ms; p < 0.001, Wilcoxon Mann-Whitney). Figure 4—figure supplement 1 has been updated.

Regarding the observation that the small sample of spikes can reveal variations in the autocorrelations, we noticed variations in the strength of the autocorrelation when we calculated it in blocks of spikes through the length recording session, but not in its lag dependency. This may reflect that autocorrelations are stronger when the rats start to sleep and may decrease in strength over time.

In addition, the new analysis (Figure 4—figure supplement 1F) indicates that many units indeed fire at the rhythm of spindle oscillations, while other units fire at longer intervals. I think that this result contradicts the authors' conclusion of "the lack of rhythmicity in individual thalamic units".

We have corrected this statement and added the new sorted thalamic units to extend and strengthen the unit analyses. We find that the peak of the autocorrelation occurred within the spindle band in 63.3 % of the curves calculated with MUA (with subsampled spikes, number matched to the units) and in 17.02 % of the units (Figure 4—figure supplement 1G shows the autocorrelation of units in which the maximum autocorrelation value falls within the spindle band lags). The autocorrelations with peak values within the spindle band had significantly higher average correlation values for MUA spikes compared to units (MUA: 0.24 ± 0.20; units: 0.05 ± 0.01; p = 0.0038, Wilcoxon Mann-Whitney). We conclude that these results suggest that most thalamic units are not strongly oscillatory and that it is the combined spikes from multiple units that underlie spindle frequency oscillations.

On the other hand, if there is an organized (grouped) sequential reactivation as the authors suggested, the cross-correlograms should reveal both co-activation and suppression at specific intervals of the spindle cycles, which did not appear in the results. Therefore, the results only suggest that thalamic neurons fire at the spindle rhythm, but not necessarily at every spindle cycle (similar to theta-cycle-skipping cells). However, these results alone do not support organized coactivation nor sequential activation as described in the authors' claims as follows: "different groups of thalamic units contribute to activity in different spindle cycles" (subsection Sparse thalamic firing and consistent activation of cell pairs during spindles”); "individual spindle cycles group and organize the sequential activation of thalamocortical ensembles" (subsection “Sparse firing of thalamic cells during spindles and its role in systems memory consolidation”); and "the sparse firing at the single-cell level will allow for the interleaved excitation of different sets of thalamocortical synapses in subsequent spindle cycles" (subsection “Sparse firing of thalamic cells during spindles and its role in systems memory consolidation”).

As pointed by the reviewer, the modulation is indeed both upward and downward, although the downward modulation tends to be smaller (maybe due to a floor effect and suppression of already low firing rates). This can be observed more clearly before normalizing by the standard deviation to obtain z-scores, as in the plot in Author response image 6, which shows the cross-correlograms after mean subtraction. The downward modulation is stronger for pairs that show stronger increased modulation and flat for pairs that do not show strong upward cross-correlation.

**Author response image 6. sa2fig6:** Cross-correlograms between spindle spikes of simultaneously recorded units in thalamus. Y-axis indicates pair ID (each row one pair). Z-axis in spikes/s (after subtracting the mean for each pair).

We have changed the sentence “individual spindle cycles group and organize the sequential activation of thalamocortical ensembles” to suggest that individual spindle cycles group and organize the activation of different thalamocortical ensembles at consistent phase lags. This is supported by the analyses of phase-difference with the added pairs of thalamic cells. We also modified the text to raise the hypothesis that the sparse firing at the single cell level could lead to the interleaved excitation of different sets of thalamocortical synapses in subsequent spindle cycles, something that can promote synaptic plasticity.

3) In the original review, the reviewers did not ask which factor, either the spindle or ripple power, contributes to the stronger modulation of thalamic activity, but the question was whether the grouping into high and low ripple power (as in the main figures) may have associated confound of spindle power differences (e.g. one group may have a higher spindle power than the other). The authors only provided the analyses in which one variable was handled independently, but they should take into account interactions between two variables, ripple and spindle power. For example, one can use a regression model of the thalamic firing modulation with both spindle and ripple power, to assess the contribution of each factor.

We addressed this comment by analyzing the modulation of thalamic cells in spindles classified based on both ripple and spindle power (results included in the text and modified panels in Figure 6E, F), and by implementing a regression model (results below). We found that the modulation of thalamic unit firing increased in spindles with high amplitude *and* high ripple power compared to spindles with low amplitude *and* low ripple power (p = 0.014, Wilcoxon signed rank test). Together with our previous analyses, the results indicate that the strongest firing modulation of thalamic units occurs with spindle amplitude alone (the largest increase in spike modulation index occurred from low to high amplitude spindles), followed by the combined effect of large spindle and ripple power, and by ripple power alone; although we did not find evidence of a significant effect of ripple power on the modulation of thalamic firing, a majority (61.29 %) of units showed a further increase in modulation index with ripple power in high amplitude spindles (average increase 18.15 ± 21.94 %), suggesting that ripple power in CA1 provides an additional boost to the spindle related modulation. We have reorganized the paragraphs in section “Spindle activation of thalamic units modulated with CA1 SWRs” for clarity and edited Figure 6 accordingly.

We used a generalized linear model (Equation 1) to regress the spikes from the thalamic units (Y; binned in 0.5s windows during sleep) on the average ripple (X_1_) and spindle (Χ_2_) power and on their multiplicative interaction (X_1_X_2_) in the same time windows. With this implementation, the ripple power parameter was significant in 87.10 % of the units, the spindle power in 90.32 % and the interaction parameter in 83.87 % (n = 31 units; significance level at 0.05, Wald test). In this model, the strength of the effect of each factor (ripple, spindle power) was stronger than their interaction; the model predicted an average 5 % decrease in thalamic spikes per unit change in ripple power, an average decrease of 2.3 % per unit of spindle power, and a weaker modulation (average 0.11 %) for the combined interaction term (an example for one thalamic unit is shown in Author response image 7). Although these models address the reviewer’s comment, they also suggest that there are additional considerations to develop valid and interpretable models. For example, the model indicates an overall decrease in firing with both ripples and spindles but does not capture the increased modulations seen in some units with peri-event histograms of detected spindles. One reason could be that the down states (K-complexes) modulate firing rate more strongly than the modulations observed within spindle cycles (Figure 3 in the manuscript). K-complexes are also associated with a rebound of activity in some thalamic units, which may confound the model further. An additional factor is that although the overall firing rate in the thalamus decreases in sleep and in association with SWRs, thalamic cells become more bursty, and modeling bursts requires the incorporation of a history dependence parameter. We think that the investigation of the interaction between sleep oscillations and thalamic firing using regressive models is an excellent continuation to the work presented here, and that it requires a dedicated project to adequately incorporate the variables we noted. The results presented in this manuscript provide helpful directions, suggesting that modeling approaches to study thalamic firing modulation in non-REM sleep should take into account ripple and spindle power as well as down-states and thalamic bursts. We include a sentence in the Discussion to point to these follow-up analyses.

log(Y) = β_0_ + β_1_X_1_ + β_2_X_2_ + β_3_ (X_1_X_2_) *Equation 1*

**Author response image 7. sa2fig7:** Example of GLM fits for the spikes of one thalamic unit as a function of ripple (left) and spindle power (right; black dots, data, jittered on the x-axis for display purposes; red, model fit). The fit is biased by the large number of data points with low or no spikes, which may be related to down-states and to pauses preceding burst firing (not included in this model).

4) In the original review, the question was not whether the mean of phase distributions changed or not by subsampling, but rather that Rayleigh's z values should be compared with the same number of spikes between groups (e.g. in their new Figure 7D).

We found that the number of spikes per SWR is not significantly different between SWRs coupled or uncoupled to spindles for units that were active in Run (p = 0.302) and for units that were not active during Run (p = 0.773; Wilcoxon signed rank). CA1 units that were active in Run were significantly more active during SWRs (coupled and uncoupled to spindles) than units not active in Run (coupled SWRs: p = 0.014; uncoupled SWRs: 0.005), but only had significantly higher Rayleigh’s z values when coupled to spindles. We have included these results in the manuscript.